# Functional Building Blocks of Neural Networks: From Network Motifs to Collective Dynamics

**Jian Zhang** [* 1 2]  **Yue Sun** [* 1]  **Wangzi Yao** [* 3 4]  **Tielin Zhang** [1 3 4 5]

## Abstract

The advancement of artificial neural networks (ANNs) has been driven by diverse and well-established architectural designs, especially in connectivity. Biological neural networks, which exhibit a rich variety of neurodynamic circuits, offer a valuable source of inspiration for developing novel ANN models. In this study, we analyze the meta-connectivity structure and introduce a network motif-based approach, in which 13 distinct motifs are modeled as functional building blocks. These motifs represent low-dimensional, fundamental components of larger network architectures. Through rigorous theoretical analysis, we classify these motifs into a three-layer hierarchical classification of their dynamical regimes and demonstrate that their hierarchical proportions critically shape collective neural dynamics. Furthermore, by embedding motif distributions into recurrent neural networks, we show that these motifs can selectively enhance either network robustness or flexibility. Collectively, our findings provide a theoretical framework—supported by extensive experiments—for understanding how specific network motifs influence the computational properties of artificial intelligence systems via their underlying dynamics. Our approach offers significant potential for analyzing and modulating neural dynamics in ANNs.

---

[*]Equal contribution [1]Center for Excellence in Brain Science and Intelligence Technology, State Key Laboratory of Brain Cognition and Brain-inspired Intelligence Technology, Institute of Neuroscience, Chinese Academy of Sciences, Shanghai 200031, China [2]University of Chinese Academy of Sciences, Beijing 100049, China [3]Institute of Automation, Chinese Academy of Sciences, Beijing 100190, China [4]School of Artificial Intelligence, University of Chinese Academy of Sciences, Beijing 100049, China [5]The Key Laboratory of Cognition and Decision Intelligence for Complex Systems, Institute of Automation, Chinese Academy of Sciences, Beijing 100190, China. Correspondence to: Tielin Zhang <zhangtielin@ion.ac.cn>.

*Proceedings of the 43rd International Conference on Machine Learning*, Seoul, South Korea. PMLR 306, 2026. Copyright 2026 by the author(s).

## 1. Introduction

Network connectivity serves as a foundational element of deep learning, as notable progress has often stemmed from innovations in network connectivity. Prominent examples include the residual connections of ResNet (He et al., 2016), the gated recurrent connections of Long Short-Term Memory (Hochreiter & Schmidhuber, 1997; Sutskever et al., 2014), and low-rank weight updates for parameter-efficient adaptation in LoRA (Hu et al., 2021). Similarly, neuroscience regards connectivity as fundamental to neural computation (Park & Friston, 2013; Bassett & Sporns, 2017). In the framework of continuous-time dynamical systems, such as continuous-time recurrent neural network (RNN) (Chen et al., 2018; Rubanova et al., 2019) and state-space models (Linderman et al., 2017; Gu et al., 2021; Gu & Dao, 2024), structured connectivity shapes rich neural dynamics (Sompolinsky et al., 1988; Wang, 2008; Mastrogiuseppe & Ostojic, 2018; Glaser et al., 2020) that not only enable complex cognitive tasks, including working memory, decision-making and motor control (Yang et al., 2019; Ni et al., 2021; Dubreuil et al., 2022; Hafner et al., 2025), but also serve as powerful approximators of real brain activity (Mante et al., 2013; Nair et al., 2023; Genkin et al., 2025). More generally, cognitive computation emerges from collective dynamics in complex networks (Fries, 2005; Mastrogiuseppe & Ostojic, 2018; Hengen & Shew, 2025), modulated by network topology and connection strength (Watts & Strogatz, 1998; Bullmore & Sporns, 2009; van Es et al., 2025).

A growing body of work affirms that connectivity, dynamics and computation are strongly interdependent in machine learning, where computation is increasingly conceptualized as state-space dynamics governed by learned connectivity (Bai et al., 2019; Orvieto et al., 2023; Sieber et al., 2024), and in neuroscience, where anatomical connectivity and synaptic coupling shape the collective dynamics that underpin cognition and behavior (Fries, 2005; Park & Friston, 2013; Thiebaut de Schotten & Forkel, 2022; van Es et al., 2025; Inácio et al., 2025). Nevertheless, we still lack a mechanistic understanding of how connectivity generates computation. Moreover, the high-dimensional nature of connectivity makes it difficult to associate it with latent dynamics, which together hinder deeper investigations into

the computational mechanisms and interpretability of deep learning. These challenges motivate us to further explore meta-connectivity structures and to distinguish connectivity patterns specifically associated with computation.

Network motifs (Milo et al., 2002), as well-established meta-connectivity structures, have been extensively studied in genetic (Shen-Orr et al., 2002; Alon, 2007), ecological (Bascompte & Melián, 2005; Stouffer et al., 2007), and other complex networks (Milo et al., 2004; Benson et al., 2016; Zhao et al., 2018). The machine learning community has begun to evaluate network motifs as structural attributes or basic building blocks of neural networks (Zhang et al., 2025; Wang et al., 2025), enabling a principled approach to characterizing network connectivity. Crucially, in contrast to the opaque, high-dimensional global connectivity, motifs exhibit intrinsically low-dimensional dynamics, enabling precise analysis (e.g., via root-locus arguments) (Prill et al., 2005). From the perspective of theoretical neuroscience, dynamics are tightly coupled to function: stable convergence confers tolerance to noise and perturbation (Kozachkov et al., 2020; Khona & Fiete, 2022), while moderate divergence supports ongoing encoding and updating of new information (Bertschinger et al., 2004; Boedecker et al., 2012). Thus, by leveraging their tractable low-dimensional dynamics, motifs offer a powerful means to bridge local structure to network-wide collective dynamics and emergent computational capabilities. However, linking motif-based dynamics to computation of network remains challenging. First, it is unknown whether local dynamical regime of motifs persist at the whole-network level and whether they reliably correspond to population network capabilities. Second, motif enumeration relies on sampling and becomes prohibitively expensive and non-scalable in deep networks with many iterative layers.

In this study, we seek to bridge motif structures and computation through their induced dynamics. To begin with, we focus on the thirteen directed 3-node motifs, viewing them as an extension of edges and as foundational basis for larger motifs. In particular, we establish analytical stability conditions to stratify the thirteen motifs into three hierarchical levels. Through a series of experiments and analyses, we reveal how motif-level dynamical regime is associated with distinct computation of neural network, with higher-stability motifs preferentially improving robustness and more weakly stable motifs supporting flexibility. Taken together, this work represents a substantial step toward a theoretical understanding of connectivity in artificial neural networks, laying a foundation for new theories and testable hypotheses about the relationship between connectivity patterns and network computation.

Our contributions are as follows:

- We analyze the dynamical regimes of the 13 motifs

and, based on stability conditions, obtain a three-layer classification result.

- By embedding motifs from different levels into the network, we respectively enhance the network's robustness and flexibility.

- We systematically examine the dynamics of motif-embedded connectivity, thereby linking the local dynamics of motifs to the collective dynamics of neural network.

**Conflict of Interest Disclosure.** The authors declare no financial conflicts of interest related to this work.

## 2. Mathematical Setup

### 2.1. Model

To investigate how structured connectivity shapes neural computation, we consider a continuous-time RNN widely used in computational neuroscience (Chen et al., 2018; Rubanova et al., 2019; Dubreuil et al., 2022). The state $\mathbf{h}(t) \in \mathbb{R}^N$ of $N$ hidden units evolves according to an ordinary differential equation (ODE) with a leak term governed by the time constant $\tau > 0$:

$$\frac{d\mathbf{h}(t)}{dt} = -\frac{\mathbf{h}(t)}{\tau} + W\phi(\mathbf{h}(t)) + \mathbf{I}(t) + \boldsymbol{\eta}(t), \quad (1)$$

where $\mathbf{I}(t)$ represents the aggregate input drive to the hidden units, defined as:

$$\mathbf{I}(t) = W_{\text{in}}\mathbf{x}(t). \quad (2)$$

The network output is given by the readout equation:

$$\mathbf{y}(t) = W_{\text{out}}\mathbf{h}(t). \quad (3)$$

In this formulation, the activation function $\phi : \mathbb{R} \to \mathbb{R}$ is $\tanh$ (applied element-wise), $W \in \mathbb{R}^{N \times N}$ is the recurrent connectivity matrix, $W_{\text{in}} \in \mathbb{R}^{N \times I}$ is the input projection matrix, $W_{\text{out}} \in \mathbb{R}^{O \times N}$ is the readout matrix, $\mathbf{x}(t) \in \mathbb{R}^I$ is the external input, $\boldsymbol{\eta}(t) \in \mathbb{R}^N$ is an additive noise process, and $\mathbf{y}(t) \in \mathbb{R}^O$ is the readout at time $t$.

For numerical simulation, we discretize the ODE using the forward Euler method with a time step $\Delta t > 0$. To simplify the analysis, we set the initial hidden state to zero, $\mathbf{h}(0) = \mathbf{0}$, and adopt a sequence-to-one setting in which the network output is taken as the readout at the final time step $T$.

### 2.2. Network Motifs and Neural Dynamics

**Definition of 3-Node motifs.** In the recurrent connectivity matrix $W$, we consider subgraphs consisting of three connected nodes $\{i, j, k\}$. There exist 13 distinct directed three-node motifs (see Fig. 1) and each motif is described

by its adjacency structure $\tilde{W}^{(m)} \in \{0, 1\}^{3\times3}$ (see Fig. A1), representing the local connectivity pattern. We focus on the distribution of these motifs within $W$ to bridge local structural attributes with network capabilities.

**Dynamics of network motifs.** To analyze the computational properties of these motifs, we associate each motif $m$ ($m \in \{1, \ldots, 13\}$) with a local circuit. To maintain consistency with the global network and account for both external stimuli and background network activity, the evolution of the internal states $\mathbf{h} = [h_1, h_2, h_3]^\top$ follows a three-node ODE:

$$\frac{dh_i}{dt} = -\frac{h_i}{\tau} + \sum_{j=1}^{3} W_{ij}^{(m)}\phi(h_j) + I_i(t) + \eta_i(t), \quad (4)$$

where $i \in \{1, 2, 3\}$ and $W^{(m)}$ is the $3 \times 3$ weight matrix defining the motif's internal coupling. The term $I_i(t)$ represents the aggregate input to the $i$-th unit. $\eta_i(t)$ denotes an additive noise process. This formulation enables a focused stability analysis of the motif as a dynamical system, characterizing its ability to maintain robust trajectories or adapt flexibly under continuous input drive.

**Stability analysis.** Following the *Hartman–Grobman Theorem*, we study the local behavior of the nonlinear system in Eq. (4) by linearizing it around an equilibrium $\mathbf{h}^*$. Let $J \in \mathbb{R}^{3\times3}$ denote the Jacobian matrix of the motif dynamics with respect to the state $\mathbf{h}$, i.e., the partial derivative $J_{ij}(\mathbf{h}^*) = \partial\dot{h}_i/\partial h_j\big|_{\mathbf{h}=\mathbf{h}^*}$. In this local analysis, the aggregate input $\mathbf{I}(t)$ and the noise process $\boldsymbol{\eta}(t)$ are treated as external perturbations. The resulting Jacobian entries are given by:

$$J_{ij}(\mathbf{h}^*) = -\frac{1}{\tau}\delta_{ij} + W_{ij}^{(m)} G_j(\mathbf{h}^*), \quad (5)$$

where $\delta_{ij}$ is the Kronecker delta and $G_j(\mathbf{h}^*) := \phi'(h_j^*)$ is the local gain at the equilibrium. This expression makes explicit how the motif connectivity $W^{(m)}$ shapes the linearized response of the state to perturbations via the spectrum of $J$. In Section 3.1, we use these stability conditions to compare the dynamical regimes of the 13 motif classes and derive a principled stratification.

### 2.3. Learning With Structural Constraints

To systematically investigate how local connectivity patterns influence computational capabilities, we treat the network's connectivity as a constrained optimization problem. The recurrent matrix $W$ is initialized with entries drawn from a Gaussian distribution. During training, the network is optimized to satisfy both task-specific requirements and structural distribution targets.

**Task-specific optimization.** For a given supervised task, the model is trained to minimize a standard task loss $\mathcal{L}_{\text{task}}$. In our framework, we employ the Mean Squared Error (MSE) to evaluate performance:

$$\mathcal{L}_{\text{task}} = \frac{1}{M} \sum_{i=1}^{M} \|\mathbf{y}_i - \hat{\mathbf{y}}_i\|^2, \quad (6)$$

where $M$ denotes the number of training samples, while $\mathbf{y}_i$ and $\hat{\mathbf{y}}_i$ represent the ground-truth labels and model predictions, respectively.

**Motif-specific regularization.** In this section, we introduce a **structural constraint** through the dedicated regularization term $\mathcal{L}_{\text{motif}} = \{\mathcal{L}_{\text{motif}}^m\}_{m=1,2,\cdots,13}$. This term encourages the network to converge toward a predefined meta-connectivity profile. The frequency $p_m$ and regularization of the $m$-th motif type are defined as:

$$p_m = \frac{c_m}{\sum_{i=1}^{13} c_i}, \quad \mathcal{L}_{\text{motif}}^m = (p_m - p_m^*)^2, \quad (7)$$

where $c_m$ (for $m = 1, \ldots, 13$) is the number of the $m$-th connected three-node motif, and $p_m^*$ is the desired target frequency.

To efficiently compute the motif regularization term while maintaining scalability, we introduce a matrix-multiplication-based expression for the exact counts of all thirteen directed three-node motif types, enabling highly parallelizable computation on modern hardware. Let $\tilde{W} \in \{0, 1\}^{N\times N}$ be the binary adjacency matrix of the connectivity matrix $W$, where $\tilde{W}_{jk} = 1$ indicates a directed connection from node $k$ to node $j$, and $\tilde{W}_{jk} = 0$ its absence (diagonal entries are zero, assuming no self-loops). The number of $m$-th motif $c_m$ can be expressed as a function $F_m(\tilde{W}; \otimes, *)$ (See appendix C), where $\otimes$ and $*$ denote Hadamard and matrix products, respectively. As we will present in Section 3.2, the non-differentiability of discrete counts is addressed through a continuous relaxation framework.

**Total objective function.** The final optimization objective integrates these two components, balancing task-driven learning with a motif-based structural prior on connectivity:

$$\mathcal{L}_{\text{total}} = \mathcal{L}_{\text{task}} + \lambda\mathcal{L}_{\text{motif}}, \quad (8)$$

where $\lambda > 0$ is a hyperparameter that governs the trade-off between task precision and structural adherence. This formulation enables us to isolate the functional contributions of different motif hierarchies by precisely controlling the topological backbone of the RNN during the learning process.

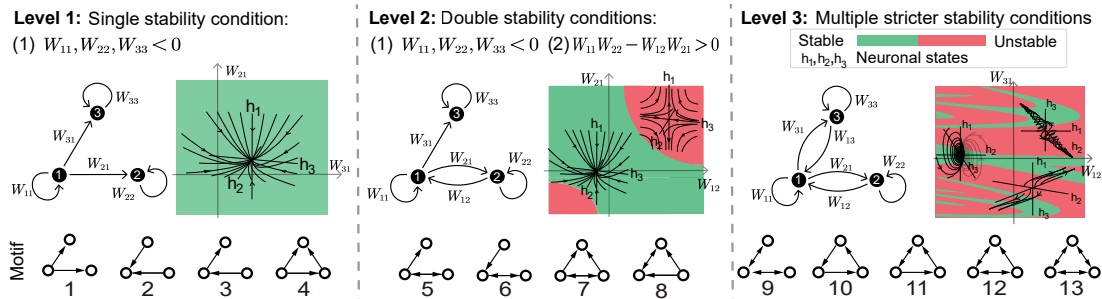

*Figure 1.* **Three-node motifs can be classified into three levels based on progressively stricter stability conditions.** In the diagrams, green regions indicate parameter regimes where the neural dynamics are stable, whereas red regions denote unstable regimes. Each panel also includes representative phase-portrait examples illustrating the corresponding dynamical behavior. Level 1 requires only negative self-connections for stability. Level 2 additionally constrains the single two-node recurrent loop. Level 3 requires further constraints, making it the strictest class.

## 3. Results

### 3.1. Stability Hierarchy of Three-Node Motifs

We characterize the stability of neural dynamics in each three-node motif by analyzing the linearization of its continuous-time system around equilibrium points. Using the *Hartman–Grobman theorem*, we approximate the local behavior of the nonlinear motif dynamics with the Jacobian evaluated at equilibrium. We then study the eigenvalues of the motif-specific recurrent submatrix $W$ to obtain sufficient stability conditions, which stratify the 13 motifs into three hierarchical levels (see Lemma 1 and Appendix B.2).

**Lemma 1 (Sufficient stability conditions).** The thirteen connected three-node motifs are categorized into a three-tiered hierarchy based on the mathematical requirements for their local stability (see Fig. 1):

- **Level-1 motifs** (IDs 1–4): The subsystem remains stable if all three nodes possess negative self-loops ($W_{11}, W_{22}, W_{33} < 0$).

- **Level-2 motifs** (IDs 5–8): Stability requires negative self-loops on all nodes ($W_{11}, W_{22}, W_{33} < 0$) **plus** an additional constraint on a specific reciprocal pair (e.g., $W_{12}W_{21} < W_{11}W_{22}$ for nodes 1 and 2).

- **Level-3 motifs** (IDs 9–13): Stability conditions are significantly more complex.

By examining the connectivity patterns corresponding to the sufficient conditions in Lemma 1, we observe that the hierarchy aligns closely with the topological complexity of recurrent structures. Level-1 motifs consist exclusively of unidirectional connections, forming directed acyclic patterns. Level-2 motifs incorporate exactly one bidirectional reciprocal pair, while Level-3 motifs feature multiple reciprocal pairs or complete three-node cycles.

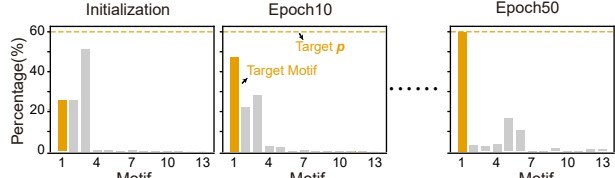

*Figure 2.* Evolution of motif distribution under structural regularization. The three panels (Initialization, Epoch 10, Epoch 50) show the percentage (y-axis) of thirteen 3-node motifs (x-axis). As training progresses, the frequency of the target motif (orange bar) increases from initialization to satisfy the predefined target threshold $p$ (dashed line)

This structural stratification reflects a fundamental transition in dynamical complexity. In a linearized system, positive self-loops ($W_{ii} > 0$) typically drive the state toward divergence (See Appendix B.1). Consequently, most stable neural systems require negative self-feedback (self-inhibition) to maintain bounded activity (May, 1972; D'Souza et al., 2023). When this prerequisite of self-stability is met, the dynamics of Level-1 motifs naturally converge to stable fixed points. In contrast, the bidirectional recurrence in Level-2 motifs elevates the stability analysis to a two-dimensional subspace, where the coupled interaction between the reciprocal pair dictates the equilibrium. Level-3 motifs further extend this to higher-order interactions, where stability cannot be reduced to simple pairwise constraints (see Fig. 1).

This hierarchy characterizes how different motif topologies support distinct local stability regimes in the corresponding three-node circuits. While our analysis is performed at the motif level, it suggests a plausible mechanism by which regulating the prevalence of these local circuits in a larger network can bias its collective dynamics and computation. In particular, enriching the network with motifs whose isolated dynamics are strongly stable is expected to promote whole-network **robustness** by increasing the prevalence of locally contracting feedback. Conversely, incorporating motifs with weaker stability can provide a structural substrate

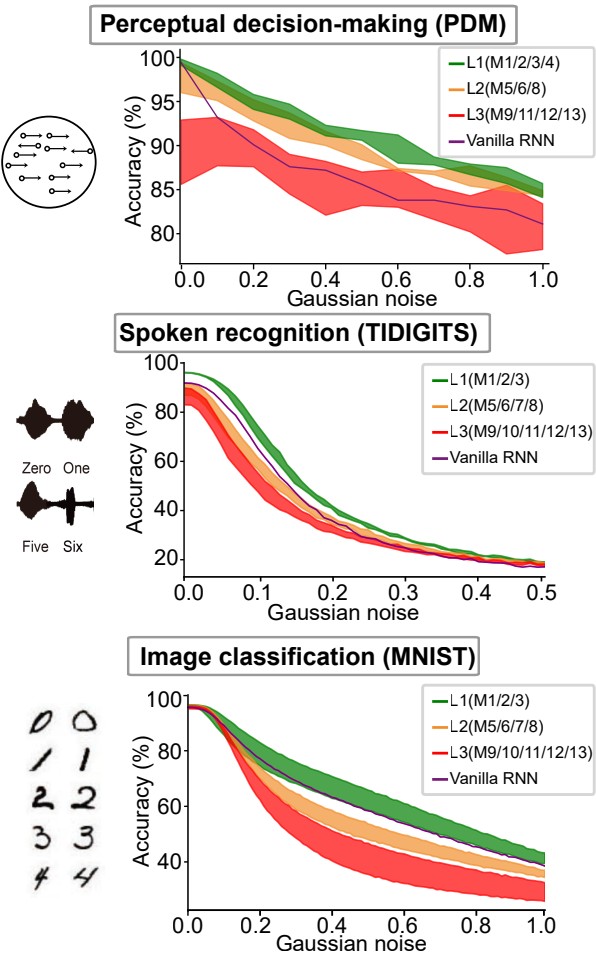

*Figure 3.* Noise-resistant recognition experiments on perceptual decision-making (PDM) (top), TIDIGITS (middle) and MNIST (bottom). The panels display classification accuracy (y-axis) against increasing Gaussian noise (x-axis). Across all three tasks, Level-1 motifs (green) maintain higher accuracy compared to the Vanilla RNN (purple), Level-2 (orange), and Level-3 (red) motifs, reflecting a clear stratification of noise tolerance based on motif stability levels.

for **flexibility**: such motifs tend to admit dynamics closer to the stability boundary, which may facilitate rapid state transitions and rich information processing when embedded in a high-dimensional network.

### 3.2. Motif Calculation and Differentiable Training

During gradient-based optimization, the discrete nature of $\tilde{W}$ presents a challenge for backpropagation. We introduce a continuous relaxation by substituting the binary $\tilde{W}$ with a differentiable proxy: $\tilde{W} \approx \sigma(\beta((W \odot W) - \theta))$, where $\sigma(\cdot)$ is the sigmoid function and $\beta$ is a temperature parameter controlling the sharpness of the relaxation and $\theta$ defines the effective connection threshold in the relaxed adjacency matrix. This relaxed adjacency is used only for motif counting and regularization, while $W$ is the recurrent weight matrix

*Table 1.* Mean accuracy averaged across all noise levels.

| | Motif | PDM Mean Acc. | TIDIGITS Mean Acc. | MNIST Mean Acc. |
|---|---|---|---|---|
| Level 1 | M1 | **91.9±1.4** | **45.0±1.9** | **67.0±3.4** |
| | M2 | 91.2±0.8 | **46.0±1.6** | 66.4±4.1 |
| | M3 | 91.2±1.2 | 44.7±0.8 | 61.3±9.2 |
| | M4 | 91.2±1.3 | 38.3±1.4 | 54.8±3.1 |
| Level 2 | M5 | 89.8±2.0 | 40.4±2.0 | 58.8±2.9 |
| | M6 | 90.9±0.8 | 41.2±2.6 | 55.6±2.9 |
| | M7 | 91.2±0.6 | 39.0±2.9 | 54.8±4.9 |
| | M8 | 89.6±1.1 | 39.0±4.1 | 50.5±5.1 |
| Level 3 | M9 | 87.5±0.7 | 38.4±2.8 | 53.5±1.8 |
| | M10 | 89.7±0.8 | 36.8±1.5 | 51.5±5.4 |
| | M11 | 87.4±1.4 | 35.8±1.9 | 50.2±3.1 |
| | M12 | 82.9±2.2 | 35.8±2.4 | 46.9±5.3 |
| | M13 | 85.9±0.8 | 43.8±1.6 | **69.4±2.8** |
| Vanilla RNN | | 87.1±1.7 | 40.9±2.5 | 61.7±2.3 |

used in the forward dynamics. This renders $\mathcal{L}_{\text{motif}}$ differentiable, enabling targeted motif distributions to be reliably embedded into the network. As shown in Fig. 2, this regularization allows the structural distribution to stabilize early and persist throughout task-specific training.

In the following section, we transition from this structural optimization framework to an evaluation of network-level computational capabilities, establishing the link between local connectivity and population behavior.

### 3.3. Computational Capabilities of Motif-Endowed Networks

To test the functional implications of our stability hierarchy, we evaluate networks endowed with specific-level motifs across two task categories with opposing dynamical requirements: **noise-resistant recognition** and **continual motor control**. We posit that these tasks demand distinct capabilities in stability and flexibility. Recognition and classification under noise require the network to suppress stochastic fluctuations. This necessitates high dynamical stability to ensure that sensory perturbations do not derail the decision-making process, a property we hypothesize is provided by the convergent dynamics of Level-1 motifs. Conversely, reinforcement learning tasks require the system to remain highly sensitive to environmental feedback to facilitate exploration and rapid action updates. This necessitates divergent-prone dynamics, which we hypothesize are facilitated by the complex feedback loops of Level-3 motifs.

We first evaluated the networks on PDM, auditory (TIDIGITS), and visual (MNIST) categorization tasks across various input noise variances. Consistent with our hypothesis, a clear performance stratification emerged: Level-1 motif-enriched networks achieved the highest classification reliability, followed by Level-2, Level-3 and vanilla. Quantitatively, networks with Level-1 motifs outperformed those with Level-2 motifs in mean accuracy by $1.0 \pm 0.4\%$

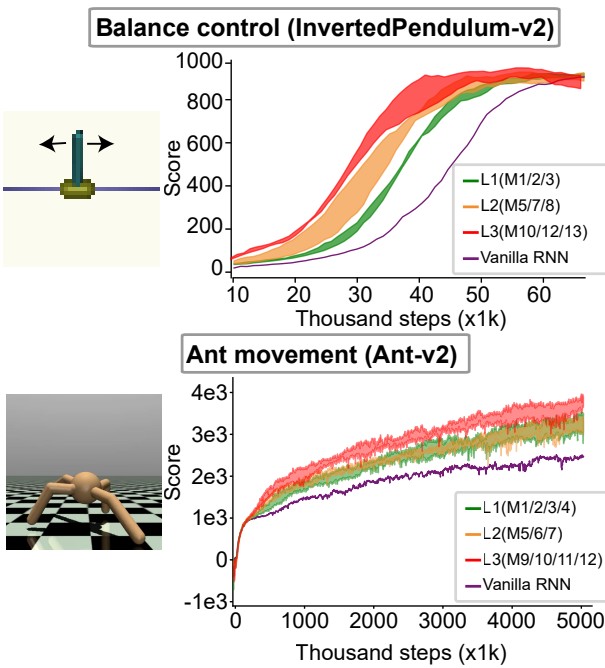

**Figure 4.** Learning curves on Inverted pendulum-v2 (top) and Ant-v2 (bottom). The plots track agent Score (y-axis) over Thousand steps (x-axis) across different motif levels. Notably, Level-3 motifs (red) exhibit a faster initial learning rate compared to Level-1 (green), Level-2 (orange), and the Vanilla RNN (purple).

*Table 2.* MuJoCo reinforcement learning performance.

| | Motif | InvertedPendulum-v2 | Ant-v2 | |
| | | Score@40k steps | Score@2m steps | Score@5m steps |
|---|---|---|---|---|
| Level 1 | M1 | $907.1 \pm 193.3$ | $2405.9 \pm 165.2$ | $3921.4 \pm 210.2$ |
| | M2 | $876.5 \pm 186.5$ | $2343.7 \pm 183.9$ | $3922.2 \pm 257.0$ |
| | M3 | $855.0 \pm 206.3$ | $2340.0 \pm 88.4$ | $3984.3 \pm 92.6$ |
| | M4 | $959.2 \pm 83.7$ | $2439.6 \pm 119.8$ | $3446.7 \pm 225.7$ |
| Level 2 | M5 | $819.3 \pm 285.5$ | $2413.2 \pm 175.7$ | $3451.5 \pm 260.5$ |
| | M6 | $962.1 \pm 113.7$ | $2548.9 \pm 206.9$ | $3902.1 \pm 259.8$ |
| | M7 | $\mathbf{984.3 \pm 47.1}$ | $2543.8 \pm 116.7$ | $3385.3 \pm 164.2$ |
| | M8 | $869.5 \pm 205.3$ | $2701.3 \pm 126.7$ | $4004.0 \pm 151.9$ |
| Level 3 | M9 | $869.5 \pm 205.3$ | $\mathbf{2893.0 \pm 116.6}$ | $3913.3 \pm 154.4$ |
| | M10 | $970.9 \pm 87.3$ | $2638.5 \pm 125.8$ | $\mathbf{4019.5 \pm 184.5}$ |
| | M11 | $\mathbf{1000.0 \pm 0.0}$ | $2567.7 \pm 230.2$ | $3918.2 \pm 214.6$ |
| | M12 | $\underline{\mathbf{1000.0 \pm 0.0}}$ | $\mathbf{2725.0 \pm 233.5}$ | $\mathbf{4258.0 \pm 166.6}$ |
| | M13 | $962.1 \pm 113.7$ | $\underline{\mathbf{3010.0 \pm 189.1}}$ | $\mathbf{4174.9 \pm 204.2}$ |
| Vanilla RNN | | $607.0 \pm 299.3$ | $1961.1 \pm 133.8$ | $2882.1 \pm 243.9$ |

theoretical framework. The specialization in computational capabilities observed, where Level-1 motifs optimize for noise robustness (see Fig. 3 and Fig. A6) while Level-3 motifs facilitate adaptive exploration(see Fig. 4 and Fig. A6), supports motif-level regularization as a way to design structural priors in ANNs. The high similarity in performance among motifs within the same stability level further supports our hierarchical classification based on intrinsic stability conditions.

**Additional Baseline Comparisons.** To further test whether the observed robustness–flexibility specialization is specific to motif-level structural priors rather than a consequence of stronger recurrent architectures or generic stability-oriented regularization, we compared motif-regularized RNNs with additional recurrent and structural baselines. The recurrent baselines include Long Short-Term Memory (LSTM) (Hochreiter & Schmidhuber, 1997) and Gated Recurrent Unit (GRU) (Cho et al., 2014), which introduce gated recurrence for sequence processing. The structural baselines include orthogonal initialization (Saxe et al., 2014), low-rank recurrent factorization (Kuchaiev & Ginsburg, 2017), and spectral normalization (Miyato et al., 2018), which respectively control initialization-induced stability, low-rank parameterization, and spectral properties of recurrent weights.

Consistent with the main results, the additional baselines do not eliminate the advantage of Level-1 motif-regularized networks on noise-resistant recognition tasks. In PDM, Level-1 networks achieved $91.9 \pm 1.4\%$, outperforming the vanilla RNN ($87.1 \pm 1.7\%$, $P < 0.001$), GRU ($86.5 \pm 7.2\%$, $P < 0.05$), low-rank factorization ($87.3 \pm 5.3\%$, $P < 0.05$), and spectral normalization ($83.0 \pm 4.6\%$, $P < 0.001$). In TIDIGITS, Level-1 networks achieved $45.0 \pm 1.9\%$, outperforming the vanilla RNN ($40.9 \pm 2.5\%$, $P < 0.001$). In Sequential MNIST, LSTM achieved the highest accuracy among the tested baselines ($68.19 \pm 1.51\%$), while

($P < 0.05$) for PDM, $5.3 \pm 0.6\%$ ($P < 0.001$) for TIDIG-ITS, and $9.5 \pm 1.2\%$ ($P < 0.001$) for MNIST, and more than networks with Level-3 motifs, by $4.7 \pm 0.6\%$ ($P < 0.001$) in PDM (Tab. 1 and Fig. 3), $5.3 \pm 0.7\%$ ($P < 0.001$) in TIDIG-ITS, and $9.0 \pm 1.2\%$ ($P < 0.001$) in MNIST. These results indicate that the convergent dynamics of Level-1 structures effectively filter out stochastic perturbations, maintaining stable categorical representations across diverse sensory modalities.

In contrast, we assessed the models on reinforcement learning tasks requiring continuous adaptation. In the Inverted Pendulum task, Level-3 motif-enriched networks demonstrated superior motor learning capacity, achieving significantly faster convergence ($18.8 \pm 6.3\%$ faster than Level-2, $P < 0.001$, and $31.8 \pm 9.2\%$ faster than Level-1, $P < 0.001$). Similarly, in the Ant-v2 environment, Level-3 motifs consistently attained higher cumulative returns across training stages. Specifically, at the 2M-step mark, Level-3 networks exceeded Level-1 scores by $384.5 \pm 125.1$ ($P < 0.05$). This advantage persisted at 5M steps, where Level-3 networks maintained a performance lead of $385.3 \pm 156.1$ ($P < 0.05$) over Level-2 and $417.5 \pm 164.3$ ($P < 0.05$) over Level-1 (Tab. 2 and Fig. 4). The sensitivity to environmental feedback afforded by the divergent-prone dynamics of Level-3 motifs is associated with efficient exploration and rapid adaptation in high-dimensional state spaces.

Collectively, these results provide empirical support for our

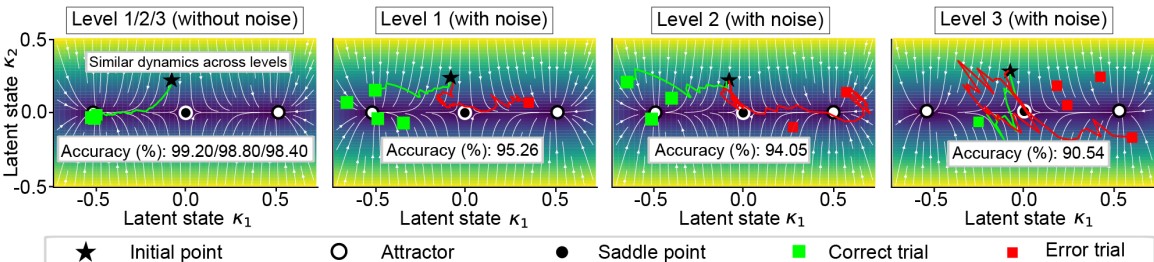

*Figure 5.* Comparison of latent state trajectories during the PDM task for Level-1, level-2 and Level-3 enriched networks. At low noise levels, both networks exhibit directed paths toward decision attractors; however, as noise variance increases (indicated by color gradient or shaded paths), Level-1 trajectories maintain high spatial coherence, while Level-3 trajectories become significantly more dispersed.

Level-1 motif-regularized networks remained competitive ($67.0 \pm 3.4\%$). These results indicate that the robustness advantage associated with Level-1 motifs is not fully explained by gated recurrence, low-rank parameterization, or spectral control alone.

A similar pattern appears in continual motor control, where Level-3 motif-regularized networks retain strong performance in adaptive reinforcement-learning environments. In Ant-v2, Level-3 networks achieved $2725.0 \pm 233.5$ at 2M steps and $4258.0 \pm 166.6$ at 5M steps, outperforming the vanilla RNN at the same milestones ($1961.1 \pm 133.8$, $P < 0.001$; $2882.1 \pm 243.9$, $P < 0.001$). In InvertedPendulum-v2, both LSTM and Level-3 motif-regularized networks reached ceiling-level performance. We further compared training-time overhead in InvertedPendulum-v2. The motif-regularized model reached convergence in $158.61 \pm 26.55$s, faster than the vanilla RNN ($224.99 \pm 18.72$s, $P < 0.001$) and spectral normalization ($372.64 \pm 61.32$s, $P < 0.001$), and comparable to the low-rank factorization baseline ($179.04 \pm 28.28$s).

### 3.4. Linking Motif Structures to Collective Dynamics

To bridge the gap between local motif statistics and collective behavior, we investigate whether the task-specific advantages of different motif levels are preserved within the network's low-dimensional latent manifold. We employ the low-rank RNN (Mastrogiuseppe & Ostojic, 2018), which approximates high-dimensional connectivity via $R$ pairs of rank vectors: $W = \frac{1}{N} \sum_{r=1}^{R} \mathbf{m}^{(r)} \mathbf{n}^{(r)\top}$. In this formulation, $\mathbf{m}^{(r)} \in \mathbb{R}^N$ represents the $r$-th output vector that determines the direction of the recurrent flow, while $\mathbf{n}^{(r)} \in \mathbb{R}^N$ denotes the $r$-th input-selection vector that defines the basis of the latent subspace. This approach is biologically motivated and mathematically tractable, as it constrains the population activity to an $R$-dimensional subspace while maintaining the network's functional capacity (Dubreuil et al., 2022). By projecting the neural activity $\mathbf{h}(t)$ onto the basis formed by the input-selection vectors $\mathbf{n}$, we define the latent variables $\boldsymbol{\kappa}(t) = \frac{1}{N} \mathbf{n}^\top \mathbf{h}(t)$. The high-dimensional dynamics can then be reduced to the following

$R$-dimensional latent system:

$$\frac{d\boldsymbol{\kappa}(t)}{dt} = -\frac{\boldsymbol{\kappa}(t)}{\tau} + \mathbf{n}^\top \phi\Big(\mathbf{h}(t)\Big) + \mathbf{m}^\top \Big(\mathbf{I}(t) + \boldsymbol{\eta}(t)\Big). \quad (9)$$

We visualized the latent dynamics using low-rank RNN ($R = 2$) during a PDM task to examine how different motif levels respond to sensory noise. A clear contrast emerged in the trajectory distributions: in networks enriched with Level-1 motifs, the latent trajectories remain tightly clustered and converge toward stable decision points even as noise variance increases (see Fig. 5). In contrast, trajectories in Level-3 enriched networks exhibit significantly larger stochastic deviations, becoming increasingly scattered under high-noise conditions. This observation indicates that Level-1 motifs effectively buffer the latent space against external perturbations, whereas Level-3 structures lead to more noise-sensitive dynamical trajectories.

To uncover how motif structures shape computational capabilities, we analyzed the network's internal processing in the Inverted Pendulum task. We collected 200 normalized environmental observations and tracked the evolution of the output deviation across multiple iteration steps within the actor RNN (See Fig. 6 and Fig. A7). Our results indicate that the motif hierarchy is associated with the network's collective dynamical regime: Level-1 networks exhibit convergent dynamics, where output deviation rapidly decays to support stable exploitation. In contrast, Level-3 networks sustain high output deviation, manifesting divergent dynamics that facilitate exploration by remaining sensitive to environmental feedback. Notably, Level-2 motifs appear to balance these two regimes, suggesting an intermediate regime between stability and flexibility.

These results show that the computational advantages identified in Section 3.3 are accompanied by consistent changes in the geometry of the network's latent manifold. By linking motif structures to collective dynamics, we show that Level-1 motifs favor convergent dynamics associated with stability and robustness, while Level-3 motifs facilitate divergent dynamics associated with flexible exploration. This consistent mapping across scales supports motif-level regularization as

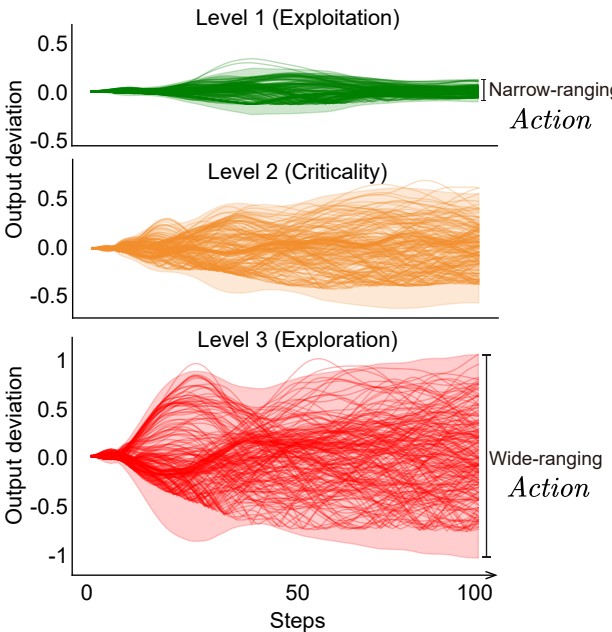

*Figure 6.* The output deviation of 200 normalized observations within the actor RNN for Level-1, Level-2, and Level-3. The x-axis represents the RNN iteration steps, and the y-axis denotes the deviation of the output results at each step. Each individual line tracks the iterative trajectory of a single observation, while the shaded regions indicate $\pm 2.56$ standard deviations. These plots clearly demonstrate that different motif levels produce distinct output ranges and deviation levels for the same set of observations, reflecting a transition from stable convergence to expansive exploration.

a method for modulating the dynamical properties of ANN to satisfy specific computational demands.

## 4. Discussion & Related Work

**Summary of Results.** In this work, we successfully integrated three-node motifs into neural networks and systematically explored the relationship between motif composition and network function. By developing a scalable, matrix-based method, we enabled the manipulation of motif distributions within RNN. Our analysis revealed that altering the abundance of different motifs impacts the network's dynamical regimes, which in turn influences network-level robustness and flexibility. These findings offer insights into how local structural features, such as motifs, govern the overall behavior of neural networks, providing a valuable tool for both theoretical and applied studies in network dynamics.

**Related Research on Motifs.** In highly evolved networks, ranging from gene regulatory networks to neural circuits, topological organization consistently exhibits patterns that deviate significantly from random Erdős–Rényi (ER) networks (Milo et al., 2002; Shen-Orr et al., 2002; Alon, 2007;

Xiong et al., 2025). It has been argued that node- and edge-level descriptions alone are insufficient to explain the complexity of such systems, as the functional properties are fundamentally shaped by structural information beyond simple pairwise interactions (Sun et al., 2023; Millán et al., 2025). Network motifs, as compact representations of this higher-order connectivity, provide a necessary framework to bridge this gap. In particular, three-node motifs serve as the basic building blocks of network architecture and offer a fundamental foundation for understanding more complex scaffolds, such as four-node motifs (Milo et al., 2002; Benson et al., 2016).

In recent years, the machine learning community has also begun to pay attention to the role of motifs, with recent work exploring how leveraging these subgraphs can improve the performance and interpretability of artificial neural networks (Zhang et al., 2025). Research in graph representation learning has increasingly focused on subgraph embedding, including methods that explicitly model substructures and their compositions (e.g., (Bar-Shalom et al., 2024; Xu et al., 2024; Zhong et al., 2025)). These approaches are often designed for graph-level prediction and may rely on enumerating or sampling subgraphs, which can be computationally expensive and difficult to scale when the number of nodes grows (Bar-Shalom et al., 2024). Simultaneously, breakthrough studies in neuroscience have mapped cortical microcircuits and large-scale connectomes with unprecedented resolution, identifying specific organizational principles such as directed and acyclic connectivity patterns (Peng et al., 2024; Xiong et al., 2025). These advancements provide a high-resolution window into the brain's complex topology and offer a biological basis for integrating structural constraints with dynamical principles in neural network design.

In contrast to existing studies that typically identify a few predominant motif types through posterior analysis, we treat the entire three-node motif distribution as a fundamental network attribute. This holistic approach enables a comprehensive examination of the network's structural hierarchy rather than focusing on isolated, high-frequency patterns. Furthermore, while most research remains descriptive or observational, our framework allows for the proactive control of network motifs. By directly incorporating motif distributions as structural priors into the recurrent connectivity, whether these distributions are derived from neural dynamics or from biological brain networks, we transition from posterior observation to the computational design of networks guided by specific computational properties.

**Neural Dynamics.** Neural dynamics represents the core engine of sequence modeling and modern recurrent architectures. In models such as structured state-space models (SSMs) and other time-series frameworks, the ability to pa-

rameterize and maintain stable dynamics is a prerequisite for long-range dependency tracking (Gu et al., 2021; Gu & Dao, 2024). Our work contributes a structural perspective to this challenge by showing that connectivity-level attributes, specifically motif distributions, directly intervene in and shape the resulting dynamical regimes.

In neuroscience, the study of neural dynamics has increasingly shifted toward the manifold perspective, where complex population activities are explained through low-dimensional latent subspaces (Mastrogiuseppe & Ostojic, 2018; Schuessler et al., 2020). Recent findings have demonstrated that specific structural configurations can induce robust dynamical objects, such as line attractors, which are essential for encoding persistent behavioral states and long-term mating dynamics (Nair et al., 2023; Vinograd et al., 2024). These biological neural networks exhibit remarkable knowledge consolidation by preserving prior information through such coordinated low-dimensional dynamics (Driscoll et al., 2017; Liu et al., 2024). Building on this view, our framework bridges local circuits and dynamical regimes by showing that three-node motifs constitute an essential local scaffold that shapes the geometry and stability of these latent dynamics.

**Computational Capability of Connectivity.** A central motivation of this work is to ground connectivity patterns in functional output by mapping local structural motifs to specific collective dynamics. Building on our motif-level stability analysis, we demonstrate that the relative abundance of three-node motifs serves as a structural attribute of networks, effectively bridging the gap between local circuits and emergent computational capabilities. Specifically, motifs that promote stable convergence are analogous to the dissipative structures that confer tolerance to noise and perturbations (Kozachkov et al., 2020; Khona & Fiete, 2022). Conversely, motifs that induce moderate divergence enlarge the range of dynamical responses, supporting the ongoing encoding and flexible updating of new information (Bertschinger et al., 2004; Boedecker et al., 2012). This link provides a principled framework to evaluate how local dynamics relate to whole-network behavior, directly addressing the challenge of structural-computational scaling. Our results suggest that motif composition can bias the robustness–flexibility trade-off in the studied settings.

## 5. Limitations and Future Directions

First, three-node motifs form the basis of more complex scaffolds, but they do not capture all higher-order interactions. Extending motif embedding and analysis to larger subgraphs (e.g., four-node motifs) is a natural next step. Second, to focus on dynamics, we studied RNN in this work. Similar topological constraints may also be important in other architectures such as MLPs and Transformers, but

a key challenge is how to define and compare dynamical properties across model classes in a principled way. Third, beyond the stable and flexible regimes, intermediate regimes may provide a trade-off between robustness and flexibility and potentially yield stronger generalizability. Characterizing these intermediate regimes, and understanding how motif composition positions a network within them, is an important direction for future study.

## Acknowledgments

This work was supported by the Brain Science and Brain-like Intelligence Technology - National Science and Technology Major Project (2025ZD0217200), Strategic Priority Research Program of the Chinese Academy of Sciences (Grant No. XDB1010302), CAS Project for Young Scientists in Basic Research (YSBR-116), Youth Innovation Promotion Association CAS, Shanghai Leading Talent Program of Eastern Talent Plan, the Lingang Laboratory Fund (Grant No. LG-GG-202402-06-07, LGL-1987-09), the Shanghai Municipal Science and Technology Project (Grant No. 25ZR1401370, 25LN3200400), Special Support Project of Guangdong Province (Grant No.0720240209). The numerical calculations in this study were carried out on the ORISE Supercomputer.

## Impact Statement

This paper presents work whose goal is to advance the field of Machine Learning. There are many potential societal consequences of our work, none which we feel must be specifically highlighted here.

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

# Appendix Contents

## A. Notation

*Table 1.* Notation used throughout the paper.

| Symbol | Description |
| --- | --- |
| $N$ | Number of recurrent units (hidden dimension). |
| $I$ | Input dimension. |
| $O$ | Output dimension. |
| $\mathbf{h}(t)$ | Continuous-time hidden state (network-level). |
| $x(t)$ | Input signal. |
| $y(t)$ | Network output (readout). |
| $W$ | Recurrent weight matrix. |
| $W_{\text{in}}$ | Input projection matrix. |
| $W_{\text{out}}$ | Output/readout matrix. |
| $\tau$ | Time constant in leaky/continuous-time dynamics. |
| $\Delta t$ | Discretization step for numerical integration (e.g., Euler). |
| $\phi(\cdot)$ | Element-wise nonlinearity; $\phi'(\cdot)$ denotes its derivative. |
| $\boldsymbol{\eta}(t)$ | Additive noise term in network hidden dynamics. |
| $\mathbf{I}(t)$ | External input drive term in hidden dynamics (network-level). |
| $\theta$ | Magnitude threshold used to binarize/relax connectivity. |
| $\beta$ | Temperature/sharpness parameter for the sigmoid relaxation. |
| $\sigma(\cdot)$ | Sigmoid function used in differentiable thresholding. |
| $\tilde{W}$ | Differentiable proxy of the binarized adjacency (used for motif counting). |
| $c_m$ | Count of the $m$-th directed 3-node motif ($m = 1, \ldots, 13$) in $\tilde{W}$. |
| $p_m$ | Empirical frequency of motif $m$ (normalized from counts). |
| $p_m^*$ | Target frequency of motif $m$. |
| $L_{\text{task}}$ | Task loss (supervised objective). |
| $L_{\text{motif}}^m$ | Motif-specific penalty term for motif $m$. |
| $L_{\text{motif}}$ | Overall motif regularizer (aggregated over motifs). |
| $\lambda$ | Weight trading off task loss and motif regularization. |
| $L_{\text{total}}$ | Total training objective. |
| $W^{(m)}$ | $3 \times 3$ internal coupling matrix for motif class $m$ (stability analysis). |
| $W_{ij}^{(m)}$ | Entry of $W^{(m)}$; directed coupling from node $j$ to node $i$ (column $j \to$ row $i$). |
| $h_i(t)$ | State of node $i$ in the 3-node motif, $i \in \{1, 2, 3\}$. |
| $I_i(t)$ | Aggregate external input drive to node $i$ (motif-level). |
| $\eta_i(t)$ | Additive noise process acting on node $i$ (motif-level). |
| $\mathbf{h}^*$ | Equilibrium (fixed point) of the motif dynamics. |
| $J$ | Jacobian matrix of the motif dynamics evaluated at $\mathbf{h}^*$. |
| $J_{ij}$ | Jacobian entry (local sensitivity of $\dot{h}_i$ w.r.t. $h_j$ at $\mathbf{h}^*$). |
| $\delta_{ij}$ | Kronecker delta. |
| $G_j$ | Local gain at equilibrium (defined by $\phi'(h_j^*)$). |
| $R$ | Rank of the low-rank recurrent connectivity (number of mode pairs). |
| $\mathbf{m}^{(r)}$ | $r$-th output (flow-defining) vector in the low-rank decomposition. |
| $\mathbf{n}^{(r)}$ | $r$-th input-selection (latent-basis) vector in the low-rank decomposition. |
| $\kappa(t)$ | Low-dimensional latent variable obtained by projecting $\mathbf{h}(t)$ onto the $\mathbf{n}$-basis. |

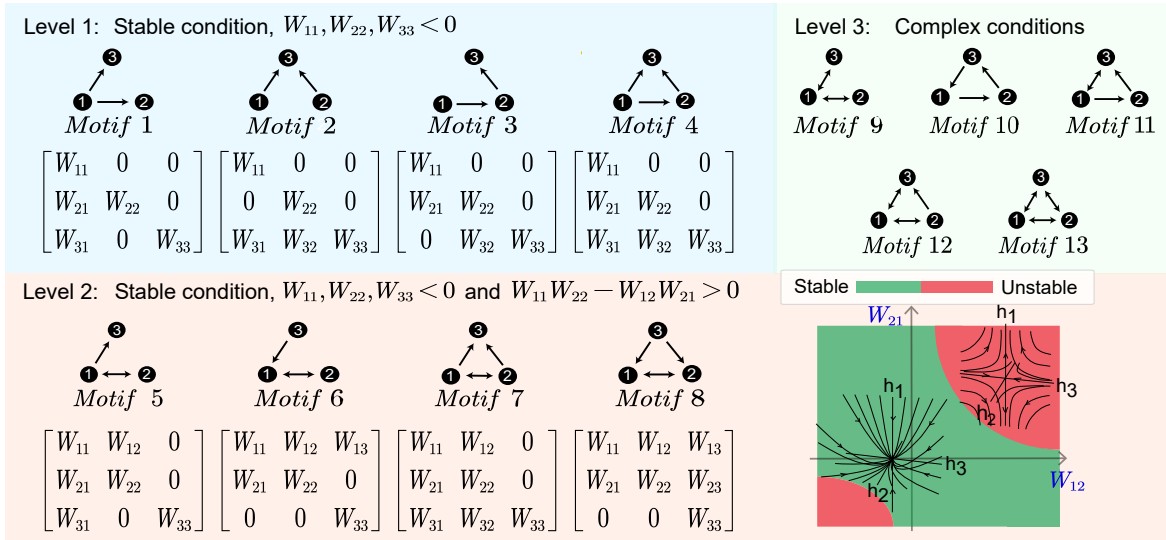

*Figure A1.* Structural atlas of 3-node motifs and their hierarchical classification. Top: Visualization of all 13 motif types organized into three levels. Middle: Representative adjacency matrices **W** for each level, where non-zero elements $W_{ij}$ denote the presence of a directed edge from node $j$ to node $i$. Bottom: Phase plane analysis illustrating the transition between stable (convergent) and unstable (divergent) regions as a function of connection strengths, providing the structural basis for the collective dynamics observed in the main text

## B. Derivation of Sufficient Stability Conditions

### B.1. Three classifications of three-node motifs in linear dynamical systems

We examine the stability of systems generated by three-node motifs through qualitative theoretical analysis of dynamical systems. Moreover, since nonlinear dynamical systems can be reduced to linear dynamical systems by the Hartman-Grobman theorem. For convenience to understand, we first consider the following motif-generated linear dynamical system:

$$\frac{dX}{dt} = \begin{bmatrix} W_{11} & W_{12} & W_{13} \\ W_{21} & W_{22} & W_{23} \\ W_{31} & W_{32} & W_{33} \end{bmatrix} X. \tag{10}$$

where $X = (x_1, x_2, x_3)$ and $\frac{dX}{dt} = \left( \frac{dx_1}{dt}, \frac{dx_2}{dt}, \frac{dx_3}{dt} \right)$. The connection matrices with motif IDs 1,2,3, and 4 can all be expressed as upper triangular matrices, and the eigenvalues of these particular connection matrices align with their respective diagonal elements.

$$W^1 = \begin{bmatrix} W_{11} & 0 & 0 \\ W_{21} & W_{22} & 0 \\ W_{31} & 0 & W_{33} \end{bmatrix}, \quad W^2 = \begin{bmatrix} W_{11} & 0 & 0 \\ 0 & W_{22} & 0 \\ W_{31} & W_{32} & W_{33} \end{bmatrix},$$

$$W^3 = \begin{bmatrix} W_{11} & 0 & 0 \\ W_{21} & W_{22} & 0 \\ 0 & W_{32} & W_{33} \end{bmatrix}, \quad W^4 = \begin{bmatrix} W_{11} & 0 & 0 \\ W_{21} & W_{22} & 0 \\ W_{31} & W_{32} & W_{33} \end{bmatrix}.$$

It implies that the eigenvalues of all four of the above matrices are $W_{11}, W_{22}, W_{33}$, and the general solution of the motif-ODEs in equation (10) is the same as the following equation:

$$X(t) = \alpha e^{W_{11}t} \begin{pmatrix} 1 \\ 0 \\ 0 \end{pmatrix} + \beta e^{W_n t} \begin{pmatrix} 0 \\ 1 \\ 0 \end{pmatrix} + \gamma e^{W_{ss}t} \begin{pmatrix} 0 \\ 0 \\ 1 \end{pmatrix},$$

where $X(0) = (\alpha, \beta, \gamma)^T$. If we assume $\max\{W_{11}, W_{22}, W_{33}\} < 0$, then $X(t) \to (0,0,0)^T$ as $t \to +\infty$. This means that as long as the diagonal is constrained to be negative, no matter what other weight values of the four motifs are, their corresponding dynamical systems are stable.

The connection matrices with motif IDs 5,6,7, and 8 can be expressed as:

$$W^5 = \begin{bmatrix} W_{11} & W_{12} & 0 \\ W_{21} & W_{22} & 0 \\ W_{31} & 0 & W_{33} \end{bmatrix}, \quad W^6 = \begin{bmatrix} W_{11} & W_{12} & W_{13} \\ W_{21} & W_{22} & 0 \\ 0 & 0 & W_{33} \end{bmatrix},$$

$$W^7 = \begin{bmatrix} W_{11} & W_{12} & 0 \\ W_{21} & W_{22} & 0 \\ W_{31} & W_{32} & W_{33} \end{bmatrix}, \quad W^8 = \begin{bmatrix} W_{11} & W_{12} & W_{13} \\ W_{21} & W_{22} & W_{23} \\ 0 & 0 & W_{33} \end{bmatrix}.$$

The characteristic equation of the above four matrices is the same as follows,

$$\det\left(\lambda I - W^i\right) = (\lambda - W_{33})\left[(\lambda - W_{11})(\lambda - W_{22}) - W_{12}W_{21}\right]$$
$$= (\lambda - W_{33})\left(\lambda^2 - (W_{11} + W_{22})\lambda + W_{11}W_{22} - W_{12}W_{21}\right) = 0,$$

for every $i \in \{5,6,7,8\}$. For convenience, define $A = \begin{bmatrix} W_{11} & W_{12} \\ W_{21} & W_{22} \end{bmatrix}$, then

$$\det\left(\lambda I - W_i\right) = (\lambda - W_{33})\left(\lambda^2 - (\operatorname{tr} A)\lambda + \det A\right) = 0.$$

Therefore, it follows from the root formula for quadratic equations that

$$\lambda_{1,2} = \frac{1}{2}\left(\operatorname{tr} A \pm \sqrt{(\operatorname{tr} A)^2 - 4\det A}\right), \quad \lambda_3 = W_{33}.$$

If we assume $\max\{W_{11}, W_{22}, W_{33}\} < 0$, then the straight-line solutions of the form $\gamma e^{W_{33}t}(0,0,1)^T$ tend to 0 as $t \to \infty$. Therefore, the three-dimensional dynamical system can be reduced to two dimensions to be studied. The dynamical classification of the planar linear ordinary differential equation can be referred to in Table 2. It follows from the qualitative theoretical analysis of dynamical systems viewpoint (Hirsch et al., 2013) that knowing $\operatorname{tr} A$ and $\det A$ tells us the eigenvalues of $A$ and therefore virtually everything about the geometry of solutions of $dX/dt = AX$. According to the sufficient condition for the stability of the dynamical system, which is that the real part of the eigenvalues is negative, it can be concluded that motifs $4, 5, 9,$ and $10$ require the additional constraints $\det A > 0$ to ensure stability.

In addition, the connection matrices with motif IDs $9, 10, 11, 12$ and $13$ are:

$$W^9 = \begin{bmatrix} W_{11} & W_{12} & W_{13} \\ W_{21} & W_{22} & 0 \\ W_{31} & 0 & W_{33} \end{bmatrix}, \quad W^{10} = \begin{bmatrix} W_{11} & 0 & W_{13} \\ W_{21} & W_{22} & 0 \\ 0 & W_{32} & W_{33} \end{bmatrix},$$

$$W^{11} = \begin{bmatrix} W_{11} & 0 & W_{13} \\ W_{21} & W_{22} & 0 \\ W_{31} & W_{32} & W_{33} \end{bmatrix}, \quad W^{12} = \begin{bmatrix} W_{11} & W_{12} & W_{13} \\ W_{21} & W_{22} & W_{23} \\ W_{31} & 0 & W_{33} \end{bmatrix}, \quad W^{13} = \begin{bmatrix} W_{11} & W_{12} & W_{13} \\ W_{21} & W_{22} & W_{23} \\ W_{31} & W_{32} & W_{33} \end{bmatrix}.$$

For convenience, we calculate the characteristic equation of motif 13, and replace the weights of the corresponding unconnected edges with 0 in the characteristic equations of other motifs:

$$\lambda^3 - (W_{11} + W_{22} + W_{33})\lambda^2 + M_2\lambda - \det W^i = 0, \quad \forall i \in \{9,10,11,12,13\},$$

where $M_2 = (W_{11}W_{22} - W_{12}W_{21}) + (W_{11}W_{33} - W_{13}W_{31}) + (W_{22}W_{33} - W_{23}W_{32})$. According to the Routh-Hurwitz stability criterion, we can obtain that the corresponding dynamical system of these motifs is stable if and only if

$$\det W < 0, \tag{11}$$
$$W_{11} + W_{22} + W_{33} < 0, \tag{12}$$
$$(W_{11} + W_{22} + W_{33})M_2 < \det W. \tag{13}$$

Unlike Level 1 or Level 2 motifs, their stability cannot be ensured solely by constraining the self-loops and a single two-node recurrent connection. As a result, we simply divide the 13 types of three-node motifs into 3 levels.

*Table 2.* **Dynamical classification of the planar linear ordinary differential equation.** From a dynamical systems point of view, the solution of the two-dimensional linear ODE $dX/dt = AX$ could be determined by the Jordan standard and eigenvalues of matrix $A$.

| Jordan standard | General solution | Eigenvalues ($\lambda_1, \lambda_2$) | Phase portraits |
|---|---|---|---|
| $\begin{bmatrix} \lambda_1 & 0 \\ 0 & \lambda_2 \end{bmatrix}$ | $X(t) = c_1 e^{\lambda_1 t}\binom{1}{0} + c_2 e^{\lambda_2 t}\binom{0}{1}$ | $\lambda_1 < 0 < \lambda_2$ 

 $\lambda_1 \le \lambda_2 < 0$ 

 $0 < \lambda_1 \le \lambda_2$ |  |
| $\begin{bmatrix} \xi & \eta \\ -\eta & \xi \end{bmatrix}$ | $X(t) = c_1 e^{\xi t}\binom{\cos \eta t}{-\sin \eta t} + c_2 e^{\xi t}\binom{\sin \eta t}{\cos \eta t}$ | $\lambda_{1,2} = \xi \pm i\eta, \xi > 0$ 

 $\lambda_{1,2} = \xi \pm i\eta, \xi < 0$ 

 $\lambda_{1,2} = \pm i\eta, \xi = 0$ |  |
| $\begin{bmatrix} \lambda & 1 \\ 0 & \lambda \end{bmatrix}$ | $X(t) = c_1 e^{\lambda_1 t}\binom{1}{0} + c_2 e^{\lambda_2 t}\binom{t}{1}$ | $\lambda_1 = \lambda_2 = \lambda < 0$ 

 $\lambda_1 = \lambda_2 = \lambda > 0$ |  |

## B.2. Three classifications of three-node motifs in nonlinear dynamical systems

In mathematics, particularly in the study of dynamical systems, the Hartman-Grobman theorem, also known as the linearization theorem, addresses the local behavior of nonlinear systems near a hyperbolic equilibrium point. The theorem asserts that the dynamics of a system in a region close to a hyperbolic equilibrium point are qualitatively similar to those of its linearization. Here, hyperbolicity means that none of the eigenvalues of the linearized system have a real part that is zero. Thus, when analyzing nonlinear dynamical systems, one can utilize the simpler linearization to study their behavior near the equilibrium point.

**Theorem.** *Hartman-Grobman theorem.* *Consider a system evolving in time with state $h(t) \in \mathbb{R}^n$ that satisfies the differential equation $dh/dt = f(h)$ for some smooth map $f : \mathbb{R}^n \to \mathbb{R}^n$. Suppose the map has a hyperbolic equilibrium state $h^* \in \mathbb{R}^n$. Let $A$ be the Jacobian matrix $[\partial f_i/\partial h_j]$ of $f$ at state $h^*$. Then the nonlinear flow is topologically equivalent to the linearized system $dX/dt = AX$ in a neighborhood of $h^*$.*

Therefore, with the help of the Hartman-Grobman theorem, the linearized system of Eq. (4) is determined by the following Jacobian matrix

$$\begin{bmatrix} -\frac{1}{\tau} + W_{11}[1 - \tanh^2(h_1^*)] & W_{12}[1 - \tanh^2(h_2^*)] & W_{13}[1 - \tanh^2(h_3^*)] \\ W_{21}[1 - \tanh^2(h_1^*)] & -\frac{1}{\tau} + W_{22}[1 - \tanh^2(h_2^*)] & W_{23}[1 - \tanh^2(h_3^*)] \\ W_{31}[1 - \tanh^2(h_1^*)] & W_{32}[1 - \tanh^2(h_2^*)] & -\frac{1}{\tau} + W_{33}[1 - \tanh^2(h_3^*)] \end{bmatrix}, \tag{14}$$

where the activation function $\phi : \mathbb{R} \to \mathbb{R}$ is $\tanh$. For convenience, we set $G_j(\mathbf{h}^*) := 1 - \left[\tanh\left(h_j^*\right)\right]^2$. Then the Jacobian matrix $J \in \mathbb{R}^{3\times3}$ of the motif dynamics with respect to the state $\mathbf{h}$, i.e., the partial derivative $J_{ij}(\mathbf{h}^*) = \partial \dot{h}_i/\partial h_j\big|_{\mathbf{h}=\mathbf{h}^*}$. The resulting Jacobian entries are given by:

$$J_{ij}(\mathbf{h}^*) = -\frac{1}{\tau}\delta_{ij} + W_{ij}^{(m)} G_j(\mathbf{h}^*), \tag{15}$$

where $\delta_{ij}$ is the Kronecker delta.

According to the analysis of the stability of linear motif dynamics, the stability of the linear dynamics is determined by the eigenvalues of the connection matrices. Noticing that, for every $x \in \mathbb{R}$, $0 < 1 - \tanh^2(x) < 1$, which implies that if we assume $max\{W_{11}, W_{22}, W_{33}\} < 0$, then the eigenvalues of the Jacobian matrices (14) with Motif IDs 1, 2, 3, and 4 are all negative, which is the same as the linear dynamical system.

When analyzing motifs with ID 5, 6, 7, and 8, Jacobian matrix (14), we define

$$A' = \begin{bmatrix} W_{11}G_1(\mathbf{h}^*) & W_{12}G_2(\mathbf{h}^*) \\ W_{21}G_1(\mathbf{h}^*) & W_{22}G_2(\mathbf{h}^*) \end{bmatrix}, \tag{16}$$

then the characteristic equation of the motifs with ID 5, 6,7, and 8 of (14) is the same as

$$[(\lambda + \frac{1}{\tau}) - W_{33}G_3(h^*)][(\lambda + \frac{1}{\tau})^2 - \operatorname{tr} A'(\lambda + \frac{1}{\tau}) + \det A'] = 0, \tag{17}$$

it implies the three eigenvalues are

$$\lambda_{1,2} = \frac{1}{2}\left(\operatorname{tr} A' \pm \sqrt{(\operatorname{tr} A')^2 - 4\det A'}\right) - \frac{1}{\tau}, \quad \lambda_3 = W_{33}G_3(h^*) - \frac{1}{\tau}. \tag{18}$$

Therefore, if we assume $max\{W_{11}, W_{22}, W_{33}\} < 0$, the stable region be expanded compared to the linear situation, and the additional constraints $W_{11}W_{22} - W_{12}W_{21} > 0$ still ensures stability.

## C. Calculating the frequency of network motifs

For convenience, we will introduce some notation that will enable us to express these calculations precisely. We aimed to determine the number of motifs consisting of three nodes in a neural network with $N$ neuron nodes. Let $\tilde{W}$ be the adjacency matrix, where $\tilde{W}_{jk} = 1$ states the presence of a directed connection from node $k$ to node $j$, $W_{jk} = 0$ states its absence. The transpose of the matrix $W$ is denoted as $W^T$, it implies that $W_{kj} = W_{jk}^T$. We define the $N_1 \times N_2$ matrix where all elements

are 1 as $\mathbf{1}_{N_1 N_2}$, and let the $N$-vector $L = \mathbf{1}_{N,1}$. It is easy to verify that if $X$ is an $N \times N$ matrix, then $L^T X L = \sum_{i,j} X_{ij}$ denotes the sum of all elements in $X$. Furthermore, to carve out unconnected edges when calculating the number of motifs, we define the $N \times N$ unit matrix $I_{N \times N}$ with diagonal 1 and the rest of the elements 0, and let $P = \mathbf{1}_{N \times N} - I_{N \times N}$. It implies that $(P - \tilde{W})_{jk} = 0$ states the presence of a directed connection from node $k$ to node $j$, and $(P - \tilde{W})_{jk} = 1$ states its absence. After ignoring self-loops of nodes, we set $\overline{W} = \tilde{W} \otimes P$ and $\dddot{W} = P - \overline{W}$.

For example, the first type of three-node motif can be calculated in the following way: the first motif with three nodes is represented as a connected graph with six edges using solid lines for connected and dashed lines for unconnected. Then for fixed $i, j, k$, only when $\overline{W}_{ji} \dddot{W}_{ij} \dddot{W}_{ik} \overline{W}_{ki} \dddot{W}_{kj} \dddot{W}_{jk} = 1$, it means that the connection topology of the three nodes $i, j$ and $k$ corresponds to the first type motif. Therefore, to count the number of this motif, considering all possible combinations of $i, j$, and $k$ within the range of 1 to $N$. In addition, when the order of nodes is swapped, specific motifs are counted repeatedly. For example, when nodes $j$ and $k$ are swapped, they represent the same first motif. Subsequently, we divided the total count by the number of duplicate cases to obtain the final count. By the method described above, the number of the first motif can be calculated as

$$c_1(W) = \frac{1}{2} \sum_{i,j,k} \overline{W}_{ji} \dddot{W}_{ij} \dddot{W}_{ik} \overline{W}_{ki} \dddot{W}_{kj} \dddot{W}_{jk}$$

$$= \frac{1}{2} \sum_{i,j,k} \overline{W}_{ji} \dddot{W}_{ji}^T \dddot{W}_{ik} \overline{W}_{ik}^T \dddot{W}_{jk}^T \dddot{W}_{jk}$$

$$= \frac{1}{2} \sum_{i,j,k} [\overline{W} \otimes \dddot{W}^T]_{ji} [\dddot{W} \otimes \overline{W}^T]_{ik} [\dddot{W}^T \otimes \dddot{W}]_{jk}$$

$$= \frac{1}{2} \sum_{j,k} \left\{ \sum_i [\overline{W} \otimes \dddot{W}^T]_{ji} [\dddot{W} \otimes \overline{W}^T]_{ik} \right\} [\dddot{W}^T \otimes \dddot{W}]_{jk}$$

$$= \frac{1}{2} \sum_{j,k} [(\overline{W} \otimes \dddot{W}^T)(\dddot{W} \otimes \overline{W}^T)]_{jk} [\dddot{W}^T \otimes \dddot{W}]_{jk}$$

$$= \frac{1}{2} \sum_{j,k} [(\overline{W} \otimes \dddot{W}^T)(\dddot{W} \otimes \overline{W}^T) \otimes \dddot{W}^T \otimes \dddot{W}]_{jk}$$

$$= \frac{1}{2} L^T [(\overline{W} \otimes \dddot{W}^T)(\dddot{W} \otimes \overline{W}^T) \otimes \dddot{W}^T \otimes \dddot{W}] L,$$

where $\otimes$ is the Hadamard product. The counting formulas for the other twelve three-point motifs can be obtained similarly.

*Table 3.* Model parameters for benchmark tasks.

| Hyperparameter | PDM | TIDIGITS | MNIST | InvertedPendulum-v2 | Ant-v2 |
|---|---|---|---|---|---|
| Input dimension | 1 | 20 | 28 | 4 | 27 |
| Hidden dimension | 512 | 512 | 512 | 512 | 512 |
| Output dimension | 1 | 10 | 10 | 1 | 8 |
| Batch size | 64 | 16 | 256 | 64 | 64 |
| Learning rate | 1e-3 | 1e-2 | 1e-3 | 3e-5 | 3e-5 |
| Motif learning rate | 1e-2 | 1e-2 | 1e-2 | 1e-2 | 1e-2 |
| Training epochs | 50 | 150 | 5 | 150 | 5000 |

## D. Conclusion

In this work, we stratified 13 three-node motifs into three-level classification by their local dynamical stability and embedded them into recurrent neural network connectivity using differentiable regularization. The experiments revealed that higher-stability Level-1 motifs exhibit superior robustness to perturbations under noise, whereas lower-stability Level-3 motifs demonstrate greater flexibility, enabling faster learning and more effective exploration. These findings establish a direct, mechanistic connection between local motif dynamical regimes and computational capabilities in artificial neural

networks, providing a bio-inspired and theoretically grounded framework for controlling neural dynamics through structured connectivity.

## E. Experimental Settings

We evaluate our motif-embedded continuous-time RNNs in two computational domains, namely noise-resistant recognition and continual motor control. All reported quantitative classification results are aggregated over 10 independent random seeds. Tables report mean $\pm$ standard deviation, while statistical tests in the text use standard error. Code is available at https://github.com/jian-prunus-prunus/Functional-building-blocks-of-neural-networks-. The robustness domain includes PDM, Sequential MNIST, and spoken recognition on TIDIGITS, while the reinforcement learning domain includes the InvertedPendulum-v2 and Ant-v2 environments. The overall training procedure is summarized in Algorithm 1, and the task-specific model dimensions and optimization hyperparameters are summarized in Table 3. The models are optimized with a joint objective that combines the task loss and the motif-based structural prior. Unless otherwise stated, the main experiments use a fixed trade-off weight $\lambda = 10^6$. The motif regularization uses an amplitude of 100 and a bias of 0.05. All thirteen motifs share the same predefined target frequency of 0.3 in the main experiments. To clarify the robustness of these choices, we provide a dedicated hyperparameter sensitivity analysis in the next section, where we further study the effects of $\lambda$, target frequency, and the relaxation parameters $\beta$ and $\theta$.

For the robustness tasks, the continuous-time ODE dynamics are discretized with the forward Euler method, and the hidden-state dimension is fixed to 512. Each task uses a task-specific input pipeline. In Sequential MNIST, each image is fed row by row as a sequence of 28 steps with 28-dimensional inputs. In TIDIGITS, each spoken utterance is first converted into Mel-frequency cepstral coefficient (MFCC) features and then reshaped to a 20 by 20 representation. To probe temporal retention, the recurrent state is further updated with 5 zero-input recurrent steps after each real input step. We use the Adam optimizer for all robustness tasks. To evaluate dynamical stability, these tasks are tested under different levels of additive Gaussian input noise.

For the continual motor control tasks, we use Proximal Policy Optimization (PPO) with recurrent actor and critic networks. The actor outputs the mean and standard deviation of a Gaussian policy for continuous control. The critic network topology is jointly constrained by the motif regularization together with the standard PPO updates. PPO uses the Adam optimizer with a rollout horizon of 2048, one update epoch per iteration, and a maximum episode length of 10000. We set the discount factor $\gamma$ to 0.98, the generalized advantage estimation (GAE) parameter to 0.98, and the clipping coefficient $\epsilon$ to 0.2. The critic network uses a weight decay of 0.001. Running-state normalization uses a denominator epsilon of $10^{-8}$ and is clipped to the range from -5 to 5. We evaluate motor learning by tracking cumulative returns and convergence speed during training. Performance is recorded at milestones including 40k, 2M, and 5M steps.

---

**Algorithm 1** Motif-Regularized Continuous-Time RNN Training

---

**Require:** Training dataset $\mathcal{D}$, target motif frequencies $\{p_m^*\}_{m=1}^{13}$, trade-off weight $\lambda$, relaxation parameters $\beta$ and $\theta$
1: Initialize recurrent weight matrix $W$, input projection matrix $W_{in}$, and readout matrix $W_{out}$
2: **for** each training mini-batch $(x, y) \in \mathcal{D}$ **do**
3:     Compute task predictions $\hat{y}$ through the ODE dynamics and evaluate $\mathcal{L}_{task} = \frac{1}{M} \sum_{i=1}^{M} \|y_i - \hat{y}_i\|^2$
4:     Compute the differentiable proxy $\tilde{W} = \sigma(\beta((W \odot W) - \theta))$ and motif counts $c_m = F_m(\tilde{W}, \otimes, *)$
5:     Evaluate the motif regularization penalty $\mathcal{L}_{motif} = \sum_{m=1}^{13} \left( \frac{c_m}{\sum_{j=1}^{13} c_j} - p_m^* \right)^2$
6:     Update $W$, $W_{in}$, and $W_{out}$ by minimizing $\mathcal{L}_{total} = \mathcal{L}_{task} + \lambda \mathcal{L}_{motif}$
7: **end for**
8: **Return** optimized network parameters $W$, $W_{in}$, and $W_{out}$

---

## F. Sensitivity Analysis

We perform a comprehensive sensitivity analysis to systematically evaluate our hyperparameter choices against three primary objectives. First, we aim to identify a parameter space with high task accuracy. Second, we seek to determine the boundary conditions for effective motif training where the network successfully matches the targeted structural profile. Third, we investigate how the learned motif distribution is stably preserved during subsequent task optimization. To address the first two objectives, we evaluate two critical parameter groups using grid search heatmaps. The first group comprises the

regularization coefficient $\lambda$ and the target motif frequency. The second group includes the continuous relaxation parameter $\beta$ and the threshold bias. To address the third objective, we analyze the training dynamics over time to observe the transition between structural constraints and task learning.

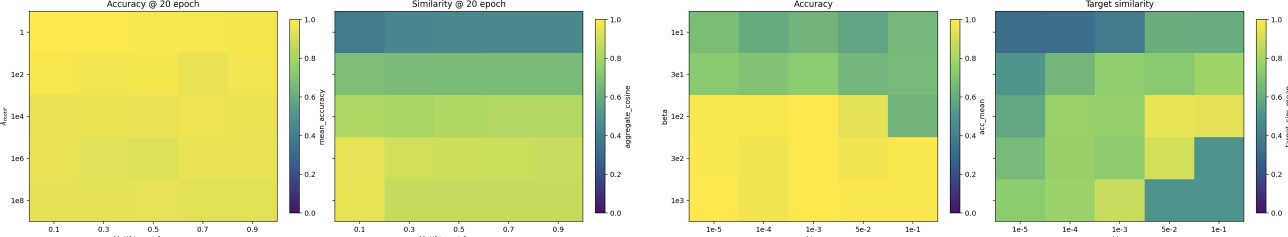

*Figure A2.* **Sensitivity analysis of motif regularization hyperparameters on the PDM task.** The left panel displays task accuracy and target motif similarity evaluated at epoch 20 across varying regularization coefficients $\lambda$ and target frequencies. The right panel displays the final task accuracy and target similarity across different relaxation parameters $\beta$ and threshold biases. These heatmaps demonstrate that high accuracy and effective motif structural matching can be achieved simultaneously within a broad and stable hyperparameter regime.

As illustrated in Figure A2, our model performance is not dependent on a narrow set of isolated optimal values. Instead, a broad parameter regime satisfies both high predictive accuracy and effective motif training. The left panel shows that task accuracy remains uniformly high across nearly all tested values of $\lambda$ and target frequencies. However, successful motif training requires a sufficiently strong regularization coefficient. When $\lambda$ is set to $10^4$ or higher, the network reliably achieves high target similarity. The right panel reveals the impact of the continuous relaxation parameters on both accuracy and structural alignment. A very small $\beta$ value produces an overly smooth proxy that degrades both task accuracy and motif similarity. Meanwhile, an excessively large threshold bias restricts the effective connectivity and harms structural matching. Selecting a moderate bias paired with a sufficiently large $\beta$ ensures that the network optimizes the targeted topology without sacrificing task performance. A practical starting range is $\lambda = 10^4$–$10^6$, $\beta = 10^2$–$10^3$, and $\theta = 0.01$–$0.05$; the target frequency is task dependent.

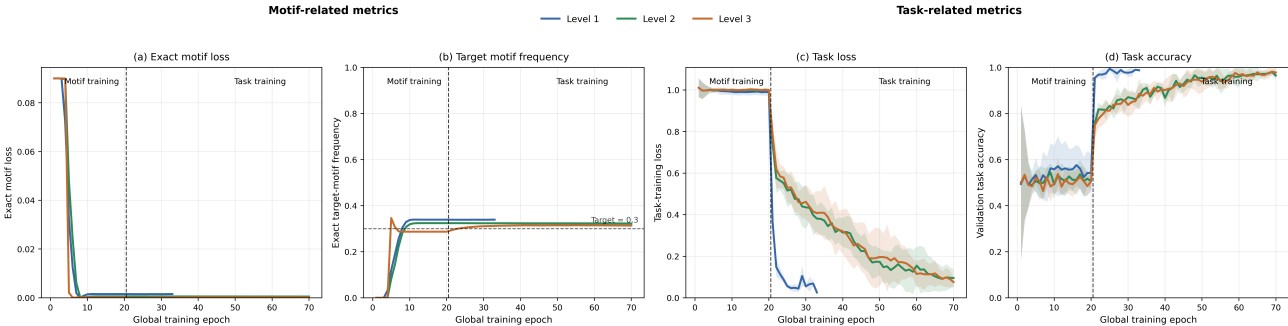

*Figure A3.* **Stability of the learned motif distribution during optimization phases.** Panels (a) and (b) track the exact motif loss and the target motif frequency. Panels (c) and (d) track the task loss and the validation task accuracy. A vertical dashed line marks the transition from motif optimization to task optimization. These trajectories show that the target motif structure is rapidly acquired and firmly retained while the network subsequently solves the specific computational task.

To address our third objective, Figure A3 demonstrates how the induced motif bias is stably preserved throughout the entire training lifecycle. During the initial motif training phase, the exact motif loss drops rapidly to zero. Concurrently, the target motif frequency converges exactly to the prescribed threshold of $0.3$. Upon transitioning to the task training phase, the network shifts its priority to the predictive objective. The task loss decreases sharply and the validation accuracy climbs steadily toward perfect classification. Crucially, the motif related metrics exhibit minimal drift during this second phase. The target motif frequency remains securely locked at the desired level. This indicates that the structural prior induced by our motif regularization is not overwritten by the task driven gradient updates. The targeted topological backbone is robustly maintained to support the underlying computational dynamics.

## G. Spectral Radius and Collective Dynamics

In this section, we examine the eigenvalue distributions of the learned recurrent matrices to observe how local circuits correlate with global spectral properties. As shown in Figure A4 and Figure A5, the spectral profiles exhibit a clear progression across the three motif stability levels. Networks dominated by Level-1 motifs display relatively concentrated eigenvalue clusters with smaller spectral radii ($\rho \sim 3.1$ to $6.3$). Moving toward Level-2 and Level-3 motifs, the spectral support broadens, resulting in a systematic increase in the spectral radius, with Level-3 reaching approximately $8.2$. Standard orthogonal and random initializations are plotted as baselines, maintaining their expected spectral radii near $1.0$. These variations indicate that different local structural constraints correspond to distinct scales in the global spectral geometry.

This stratification in spectral scale provides a descriptive reference for interpreting the collective network behaviors shown in Figure A6 and Figure A7. While downstream task dynamics are inherently non-linear, the empirical spectral radius serves as a linear proxy to understand the underlying state-space trajectories. Specifically, the tighter spectral bounds of Level-1 motifs correspond to a more contractive dynamical regime, which visually aligns with the network's capacity to suppress noise during recognition tasks. Conversely, the expanded spectral footprints of Level-3 motifs suggest an expansive regime capable of supporting the high-gain, flexible trajectories required for reinforcement learning environments. Consequently, global spectral geometry serves as a potential feature through which local circuit properties modulate these collective network dynamics.

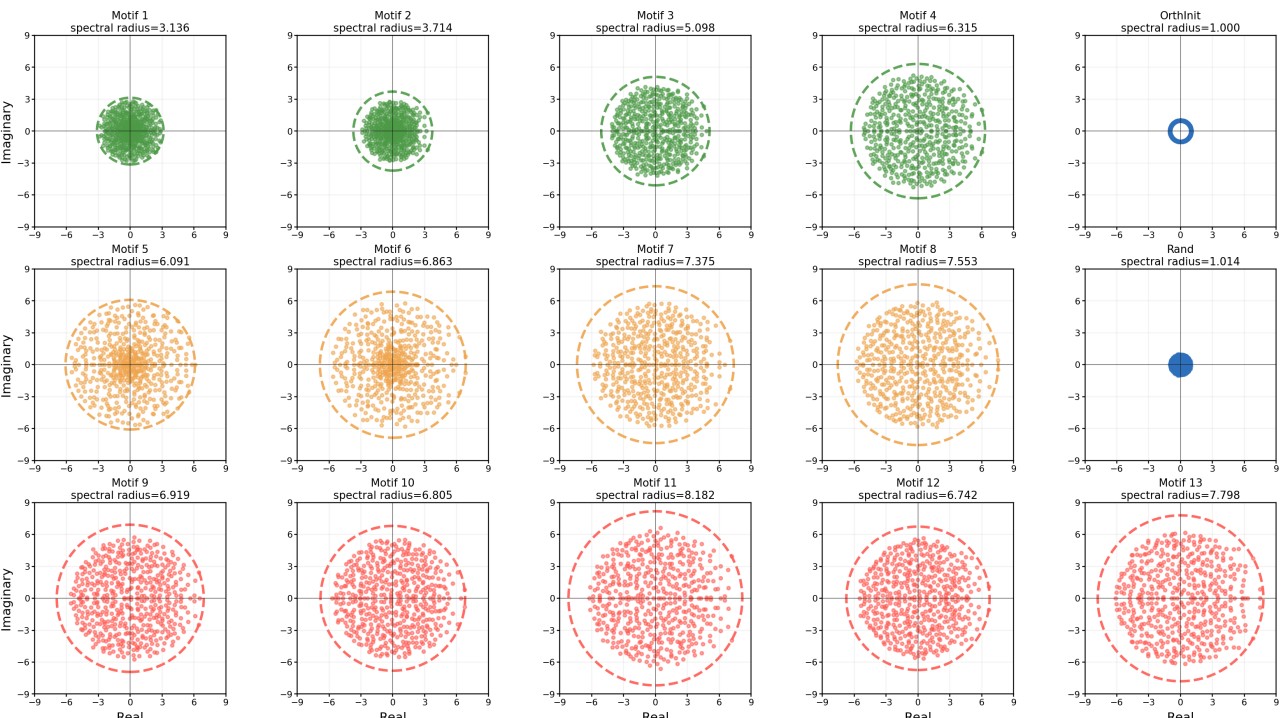

*Figure A4*. **Eigenvalue distributions of recurrent matrices under distinct motif conditions.** Each panel displays the complex plane eigenvalue distribution of the learned recurrent matrix alongside its measured spectral radius. The motifs are categorized by stability level. Level-1 is depicted in green, Level-2 in orange, and Level-3 in red. Orthogonal and random initializations are rendered in blue for baseline comparison. The learned spectra diverge significantly based on motif composition. Level-1 motifs generate densely concentrated eigenvalue clouds with notably smaller spectral radii. Level-2 and Level-3 motifs produce substantially broader spectral support and correspondingly larger radii.

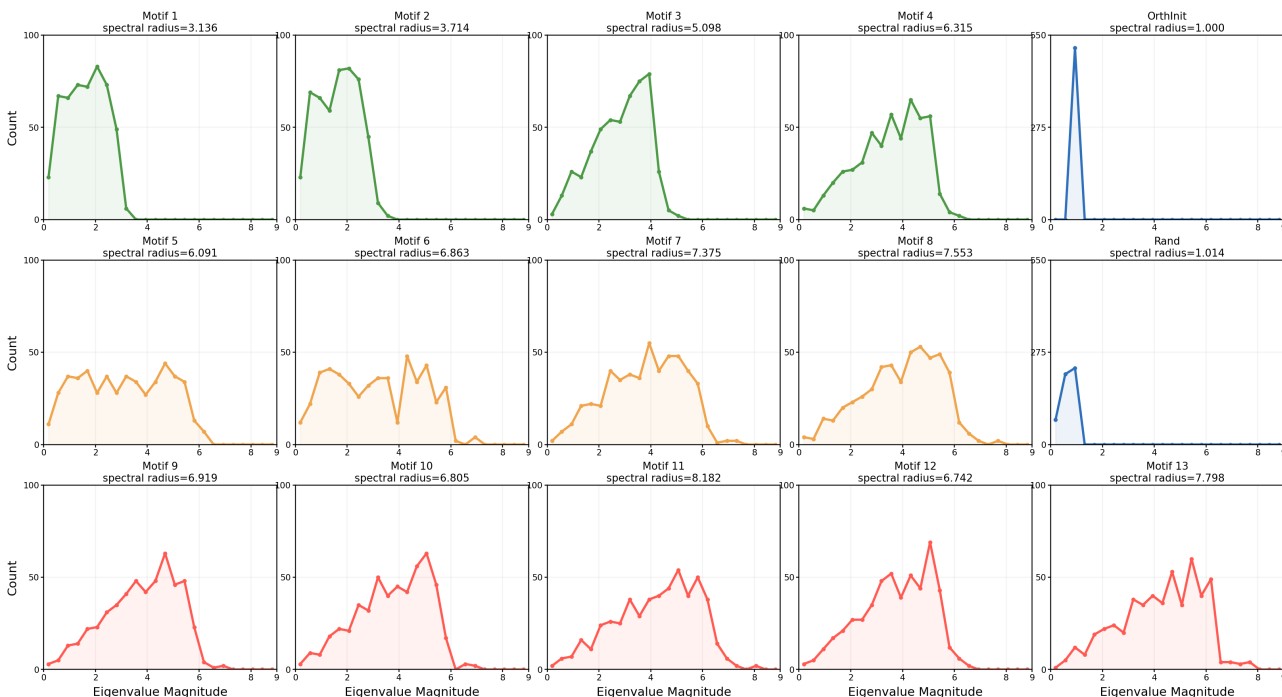

*Figure A5.* **Distribution of eigenvalue magnitudes across motif conditions.** Each panel presents the empirical probability distribution of eigenvalue magnitudes for the learned recurrent matrix alongside the specific spectral radius. Corroborating the findings in Figure A4, Level-1 motifs concentrate probability mass at significantly smaller magnitudes. The inclusion of Level-2 and Level-3 motifs systematically shifts the magnitude distribution toward larger values. This underscores how motif regularization shifts the global spectral scale of the recurrent dynamics.

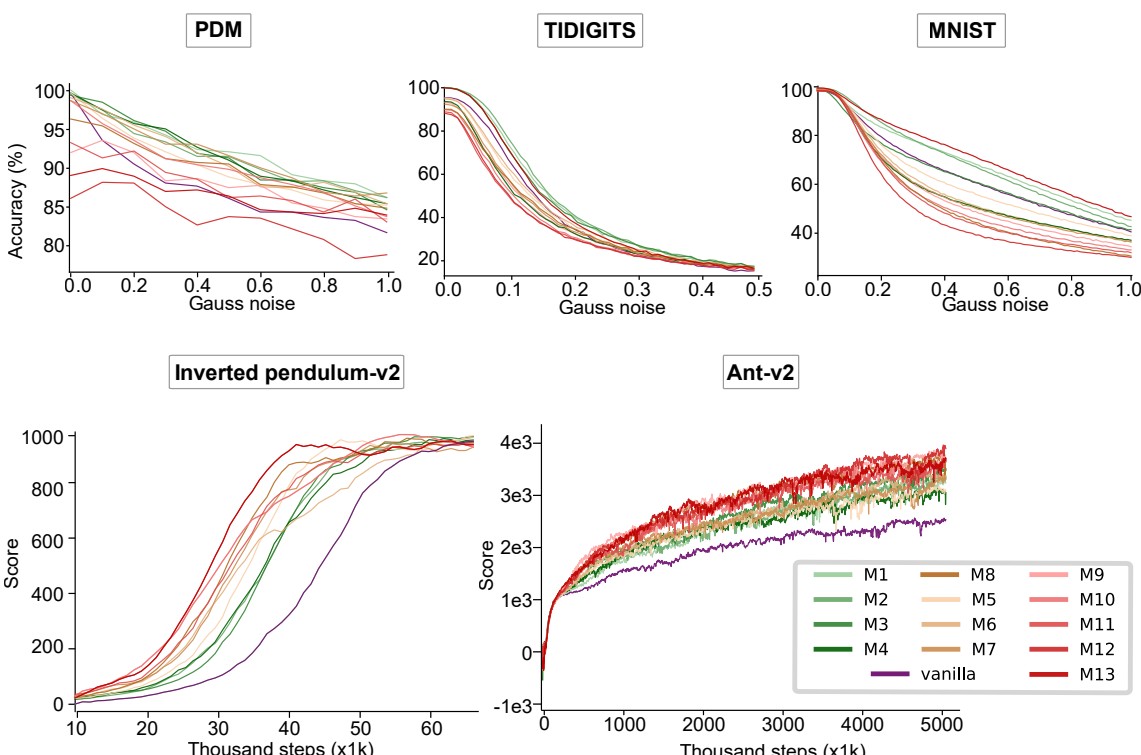

*Figure A6.* **Motif stability levels induce task dependent regimes on the stability and flexibility spectrum.** **Top (PDM/TIDIGITS/MNIST):** Classification accuracy (y-axis) is evaluated against escalating Gaussian noise variance (x-axis). Networks enriched with Level-1 motifs (green) consistently sustain higher accuracy compared to Level-2 (orange), Level-3 (red), and the Vanilla RNN (purple). This shows a stratification of noise tolerance associated with motif stability. This behavioral pattern supports our hypothesis that the locally convergent fixed point dynamics generated by Level-1 motifs filter stochastic fluctuations and stabilize categorical representations. **Bottom (InvertedPendulum-v2/Ant-v2):** Episodic return (y-axis) is recorded across training steps (x-axis). Unlike noise robust recognition, reinforcement learning demands highly flexible and feedback sensitive dynamics. Networks biased toward Level-3 motifs systematically achieve higher returns over extended horizons. This performance aligns with our theoretical proposition that feedback rich Level-3 motifs facilitate high gain and divergence prone dynamics. These dynamics amplify network responsiveness to environmental feedback and accelerate policy optimization.

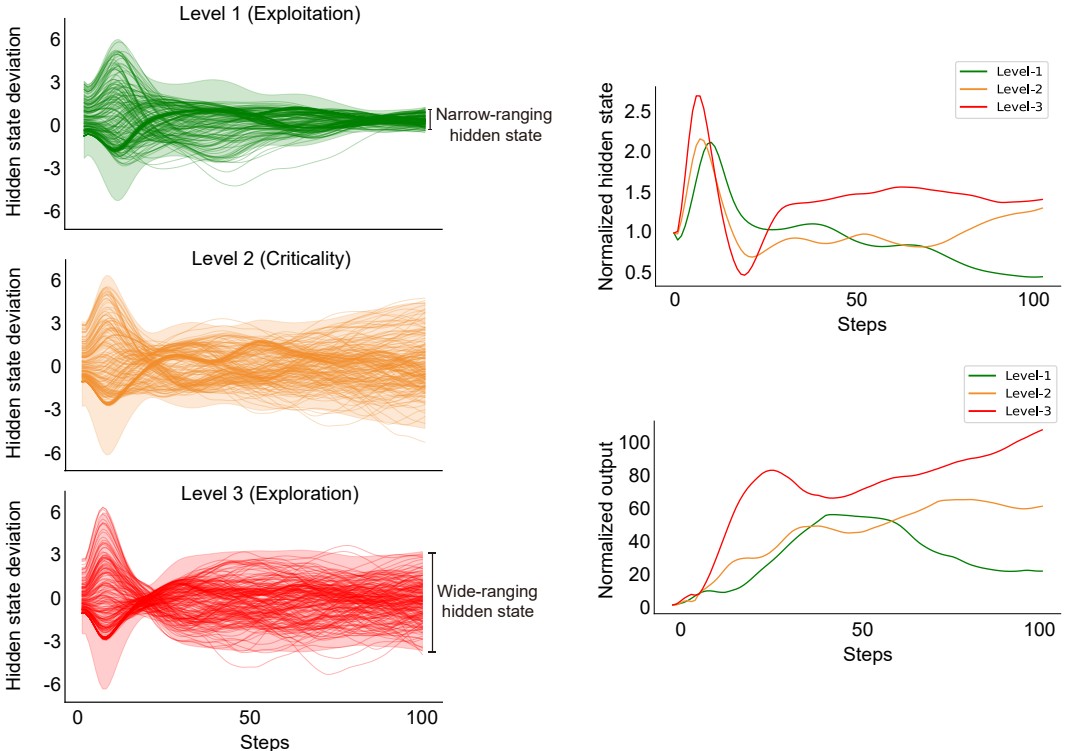

*Figure A7.* **Stability dependent internal dynamics across recurrent iterations. Left:** Per-sample trajectories of *hidden state deviation* over internal computational steps for networks biased toward Level-1 (Exploitation), Level-2 (Intermediate), and Level-3 (Exploration). Level-1 trajectories contract rapidly and remain *narrow-ranging*. This reflects locally convergent dynamics that heavily penalize deviations. Level-3 trajectories occupy a *wide-ranging* band. This behavior is consistent with high gain and exploration facilitating dynamics. Level-2 networks exhibit an intermediate dynamical regime. **Right:** Population deviation summaries of *normalized hidden state* and *normalized output* across computational steps. This plot isolates the distinct dynamical regimes induced by motif composition. The term "normalized" specifically indicates that the quantity is divided by its initial value at the **first** computational step, calculated as $z_t/z_1$.

