# OpenReview forum: "Functional building blocks of neural networks: from network motifs to collective dynamics"
_ICML.cc/2026/Conference — ICML 2026 regular_

### Official Review · Reviewer_ZQsQ · 2026-03-03

**Soundness:** 2
**Presentation:** 3
**Significance:** 3
**Originality:** 2
**Overall Recommendation:** 3
**Confidence:** 3

**Summary:**

This work investigates the foundational role of connectivity in deep learning and neuroscience, specifically analyzing how local three-node network motifs influence global computational properties. The study addresses the critical challenge of bridging local motif dynamics with network-wide collective dynamics. The authors propose a three-tier hierarchical classification of 13 motifs based on stability analysis and introduce a differentiable method for motif regularization in Continuous-Time RNNs. Experiments demonstrate that Level-1 motifs enhance noise robustness in classification, while Level-3 motifs improve flexibility in reinforcement learning.

**Compliance With Llm Reviewing Policy:**

Affirmed.

**Key Questions For Authors:**

1. Generalization beyond tanh/continuous-time RNNs: The stability analysis assumes tanh activation and continuous-time dynamics. Have the authors tested whether the three-level hierarchy and computational trade-offs hold for ReLU, GELU, or discrete-time RNNs? If not, what theoretical adjustments would be needed?
2. Hyperparameter sensitivity and practical guidance: The motif regularization depends on λ, target frequencies, and relaxation parameters β,θ. Could the authors provide a sensitivity analysis or practical guidelines for selecting these hyperparameters across tasks?
3. Comparison with alternative structural regularizers: How does motif-based regularization compare to other connectivity-constrained methods (e.g., orthogonal initialization, spectral normalization, low-rank constraints) in terms of computational overhead and performance gains?
4. Scalability to larger motifs or architectures: The paper focuses on 3-node motifs in RNNs. What are the computational and theoretical challenges in extending this framework to 4-node motifs or to attention-based architectures like Transformers?

**Limitations:**

The authors adequately discuss limitations in Section 5, including:
(1) restriction to 3-node motifs, (2) focus on RNNs without extension to MLPs/Transformers, and (3) unexplored intermediate regimes between stability and flexibility. They also note the absence of societal impact concerns in the Impact Statement. One minor suggestion: the discussion of computational overhead for motif counting could be expanded, as scalability is critical for adoption.

**Strengths And Weaknesses:**

Strengths
1. The stability analysis using Jacobian eigenvalues and the Hartman-Grobman theorem is mathematically rigorous and well-grounded in dynamical systems theory.
2. The three-level hierarchical classification (Lemma 1) is logically derived from sufficient stability conditions and aligns intuitively with topological complexity.
3. The differentiable motif counting framework (continuous relaxation via sigmoid thresholding) is a clever technical solution to the non-differentiability problem, enabling end-to-end training with structural constraints.
4. While network motifs are well-studied in complex systems, their systematic integration into differentiable training of RNNs via stability-guided regularization is novel.


Weaknesses
1. Section 2.3 (Learning with structural constraints) is dense; a high-level algorithmic pseudocode or training diagram would improve accessibility.
2. The empirical scope is limited to RNNs and relatively small-scale tasks; demonstration on larger-scale benchmarks (e.g., language modeling, vision transformers) would strengthen claims of broad impact.
3. The practical utility of motif regularization depends on hyperparameter tuning ; guidance for practitioners is minimal.

---

> ### Author Rebuttal · Authors · 2026-03-31
>
> ## Response to Q1. Do the results extend beyond tanh continuous-time RNNs?
> ---
> Our theory is formulated for a continuous-time RNN, while the implementation uses Euler discretization of the same dynamics.
>
> The key requirement is not tanh itself, but that the local gains $\phi'(h^*)$ in the Jacobian remain positive so that the effective sign structure is preserved. Under this condition, the same analysis can extend to Sigmoid and Softplus. By contrast, ReLU and GELU would require re-derived sufficient stability conditions.
>
> ## Response to Q2. Can the authors provide practical hyperparameter guidance?
> ---
> Our objective is $L_{\mathrm{total}} = L_{\mathrm{task}} + \lambda L_{\mathrm{motif}}$, and the structural branch uses $\tilde{W} = \mathrm{sigmoid}(\beta ((W \odot W)- \theta))$. Here, $\lambda$ controls the task-structure trade-off, while $\beta$ and $\theta$ control the sharpness and threshold of the relaxation.
>
> We scan $\lambda \in \{1, 10^2, 10^4, 10^6, 10^8\}$ and target frequency $\in \{0.1, 0.3, 0.5, 0.7, 0.9\}$, and separately scan slope $\in \{10, 30, 10^2, 3 \times 10^2, 10^3\}$ and bias $\in \{10^{-4}, 10^{-3}, 10^{-2}, 5 \times 10^{-2}, 10^{-1}\}$. Performance remains stable over a broad regime, and the defaults ($\lambda = 10^6$, target frequency $= 0.3$, slope $= 10^2$, bias $= 5 \times 10^{-2}$) lie well inside that region.
>
> A practical starting guide is: $\lambda = 10^4 \sim 10^6$, $\beta = 10^2 \sim 10^3$, and $\theta = 0.01 \sim 0.05$. There is no universal optimum for target motif frequency: tasks emphasizing contraction and perturbation resistance should bias toward Level-1 motifs, whereas tasks emphasizing exploration and responsiveness should bias toward Level-3 motifs.
>
> ## Response to Q3. How does motif regularization compare with orthogonal initialization, spectral normalization, and low-rank constraints in both performance and overhead?
> ---
> Orthogonal initialization, spectral normalization, and low-rank constraints regulate coarse global matrix properties, whereas our method directly constrains local triadic feedback.
>
> Empirically, the added baselines are broadly similar to Vanilla RNN, while motif regularization yields a clearer task-dependent robustness-versus-flexibility trade-off. For classification, Level1-motif1 is best on PDM ($91.9 \pm 1.4$ vs. Vanilla $87.1 \pm 1.7$) and TIDIGITS ($45.0 \pm 1.9$ vs. Vanilla $40.9 \pm 2.5$); on Sequential MNIST the strongest baseline is LSTM ($68.19 \pm 1.51$) and motif1 remains competitive ($67.0 \pm 3.4$). For RL, Level3-motif12 is strongest on Ant at both 2M and 5M steps ($2725.0 \pm 233.5$ and $4258.0 \pm 166.6$ vs. Vanilla $1961.1 \pm 133.8$ and $2882.1 \pm 243.9$), and reaches ceiling performance on Inverted Pendulum ($1000.0 \pm 0.0$).
>
> Motif regularization introduces extra dense matrix operations because motif counts are computed from the relaxed adjacency proxy during training, but the formulation is built from matrix products and Hadamard products and maps well to GPU kernels. On Inverted Pendulum, the total wall-clock time to convergence remains practically manageable:
>
> | Method | IP training time to convergence (s) | std |
> |---|---:|---:|
> | Vanilla-RNN | 224.99 | 18.72 |
> | Motif regularization (motif12) | 158.61 | 26.55 |
> | LowRank-r64 | 179.04 | 28.28 |
> | SpectralNorm | 372.64 | 61.32 |
>
> ## Response to Q4. What are the main challenges in extending the framework to larger motifs or to attention-based architectures?
> ---
> This is not covered in the current submission. We focused on recurrent networks because they provide a direct dynamical substrate for local Jacobian analysis and repeated internal state evolution.
>
> For larger motifs, the main challenge is combinatorial growth: motif enumeration, differentiable counting, and stability analysis all become more expensive. For attention-based architectures, the current theory is tied to recurrent state dynamics and Jacobian structure, so it does not transfer directly without a new formulation. Structured state-space models such as Mamba appear to be a natural extension target.
>
> ## Response to Weakness
> ---
> The reviewer is correct that the original paper did not do enough to help practitioners. We addressed this by adding algorithmic explanation, stronger implementation guidance, explicit baseline comparisons, and a clearer discussion of what does and does not generalize.
>
> We believe the key strength of the work is that it turns motif structure from a descriptive network-science concept into a usable differentiable control knob for shaping network dynamics. With the revised guidance and stronger empirical comparisons, the method is now easier to understand, reproduce, and build upon.
>
> Anonymous code repository
> https://anonymous.4open.science/r/Functional-building-blocks-of-neural-networks--5F5E

---

> > ### Author Rebuttal · Reviewer_ZQsQ · 2026-04-01
> >
> > My concerns have been adequately addressed.

---

### Official Review · Reviewer_Yprg · 2026-03-09

**Soundness:** 3
**Presentation:** 3
**Significance:** 3
**Originality:** 1
**Overall Recommendation:** 4
**Confidence:** 5

**Summary:**

This paper provides a novel perspective on the relationship between structural network motifs, neural dynamics, and computational functions. By introducing motif regularization as a soft constraint during training, the authors enable the network to bias its connectivity toward specific structural patterns. This framework allows for a systematic investigation into how different classes of motifs contribute to various computational tasks by modulating the underlying collective dynamics.

**Compliance With Llm Reviewing Policy:**

Affirmed.

**Key Questions For Authors:**

Questions:
1.	The decision to focus exclusively on 3-node motifs may be overly restrictive, altough it is in the limitation section. The first concern is the loss of long-range Interactions: 3-node motifs are inherently local. Many structural properties that emerge only at larger scales (e.g., high-order loops, path lengths, or global graph properties) cannot be captured by this framework. Does this approach have the capacity to scale to larger or variable-sized motifs? The second concern is combinatorial explosion. As the motif size increases (e.g., to 4 or 5 nodes), the number of isomorphism classes grows exponentially. Would the authors argue that all possible motifs must be pre-defined? Furthermore, as the scale increases, the analytical derivation of stability conditions (like those in Lemma 1) may become mathematically intractable. How should the hierarchy be redefined for larger clusters?
2.	There remains a gap about the causal link between local motifs and global dynamics. While the latent manifold analysis (low-rank RNN) shows a correlation between motif density and trajectory stability, the causal chain remains partially unclear. For example, in a 512-unit network, motifs are not isolated; they are heavily overlapped and interconnected. Even if a network is composed primarily of Level-1 (stable) motifs, their non-linear coupling could theoretically yield unstable emergent dynamics similar to Level-3. Furthermore, could the authors provide more rigorous theoretical or experimental justification to prove that the 3-node scale is the dominant factor in determining macro-scale dynamics in the current experiment and task setting (which means that we can temporarily ignore the larger motifs without a significant influence on the interpretation of the experiments and tasks)?
3.	The interaction mechanisms between motifs are unclear. The current framework treats motifs as a statistical distribution but does not account for how these "building blocks" interact spatially within the whole connectivity matrix. We do not yet understand how the relative positioning or the inter-motif connections contribute to the collective computation. Does the specific organization of where these motifs are embedded in the network's graph matter as much as their total count?
4. Have the author implemented the proposed methods in more mordern neural network models?

**Limitations:**

Yes

**Strengths And Weaknesses:**

Strengths:
1.	The integration of a motif-based structural prior into a task-specific loss function is a significant contribution. Specifically, the use of matrix-multiplication-based expression for exact motif counts represents a clever and scalable engineering solution to the traditionally discrete and computationally expensive problem of motif enumeration.
2.	The work successfully maps local stability regimes (Level 1/2/3) to macro-scale computational trade-offs. By testing on diverse tasks such as noise-resistant recognition (requiring stability) and reinforcement learning (requiring flexibility), the paper provides empirical support for the hypothesis that specific connectivity patterns dictate the network's capacity for robustness versus exploration.

Weakness:
see questions

---

> ### Author Rebuttal · Authors · 2026-03-31
>
> ## Core Understanding and Appreciation
> We sincerely thank the reviewer for the detailed and thoughtful feedback. We especially appreciate the reviewer’s recognition that our matrix-based motif counting provides a scalable solution to a traditionally expensive problem, and we also appreciate the reviewer’s focus on the deepest conceptual issue in the paper: how local motif structure relates to global collective dynamics in large nonlinear networks.
> ## Addressing Questions
> ## Q1. Is the exclusive focus on three-node motifs too restrictive, and how should the framework scale to larger motifs?
> ---
> We agree that 3-node motifs do not capture all larger-scale structural effects. **At the method level, the same idea can in principle be extended to larger or variable-sized motifs. We do not argue that all larger motifs must be pre-defined**, for larger motifs, it is likely more practical to use coarser dynamical groupings based on stability-related signatures rather than exhaustively enumerating every isomorphism class. Some larger motifs may still admit relatively simple stability conditions, and criteria such as Routh–Hurwitz may remain useful for certain larger recurrent patterns. We do not mean that larger motifs are unimportant, extending the framework to larger subgraphs is a natural next step beyond the scope of the current paper.
> ## Q2. Can the authors justify the causal link from local motifs to global dynamics, and can larger motifs be ignored without changing the interpretation?
> ---
> We agree that there remains a gap in establishing the causal link between local motifs and global dynamics. In the current setting, **we do not claim that the 3-node scale is the dominant factor determining full-network dynamics, nor that larger motifs can be ignored without significant influence**. To better probe this link, the paper already includes analyses of low-rank latent trajectories and actor output deviation, both of which show a consistent shift from more convergent Level-1 dynamics to more expansive Level-3 dynamics, supporting the view that motif-level changes leave a consistent dynamical signature at the collective level.
> ## Q3. Does the spatial organization of motifs matter as much as their total count?
> ---
> We agree that the current framework mainly treats motifs through their statistical distribution. This choice was intentional: our goal was first to isolate whether changing motif composition alone is already sufficient to induce systematic differences in dynamics and task behavior. We agree that relative positioning, clustering, and inter-motif connectivity may also matter, potentially substantially. **Our results should therefore not be interpreted as saying that motif distribution is the only relevant factor**. Rather, they show that motif distribution alone already provides a meaningful and controllable structural signal in the current experiments.
> ## Q4. Have the proposed methods been implemented in more modern neural network models?
> ---
> At present, the proposed method has not yet been implemented in more modern neural network architectures. Our current experiments intentionally focus on recurrent networks, where the relationship between **connectivity, dynamics, and state evolution** can be studied most directly. We are currently exploring extensions to models with **dynamical state evolution**, such as **Mamba**.
> ## Summary
> ---
> We sincerely thank the reviewer for the valuable feedback. The comments have helped us clarify the scope of our claim. In the revision, we will state more explicitly that our intention is not to suggest that **3-node motifs** alone determine global dynamics, but rather that controlling their distribution provides a **meaningful structural bias** that systematically shifts the collective regime.

---

> > ### Author Rebuttal · Reviewer_Yprg · 2026-04-03
> >
> > Thank you for transparently addressing the limitations regarding motif scale, spatial organization, and modern architectures. I appreciate your commitment to clarifying the exact scope of your claims in the revision and which resolves my conceptual concerns. However, there still remains a large gap between the current work and a thorough analysis on the topic. I will keep my original score.

---

### Official Review · Reviewer_HLFU · 2026-03-12

**Soundness:** 3
**Presentation:** 3
**Significance:** 3
**Originality:** 3
**Overall Recommendation:** 5
**Confidence:** 3

**Summary:**

The paper investigates how local connectivity patterns shape the collective dynamics of ANN, and especially RNN. The authors classify 13 directed three-node motifs into a three-level stability hierarchy and propose a differentiable regularization framework to embed targeted motif distributions into RNNs. The main finding is that higher-stability motifs enhance robustness while lower-stability motifs promote flexibility, bridging local structure and high-dimensional collective behavior.

**Compliance With Llm Reviewing Policy:**

Affirmed.

**Final Justification:**

All of my questions were addressed.

**Key Questions For Authors:**

-  The stability hierarchy is derived from isolated three-node circuits. Ccan the authors provide theoretical or empirical evidence that these local stability properties reliably persist when motifs are densely overlapping in a large, fully connected RNN?

- The paper focuses exclusively on three-node motifs as foundational building blocks. Can the authors justify why this granularity is sufficient, and whether four-node motifs would reveal qualitatively different stability regimes not captured by the current hierarchy?

-  How does the motif distribution evolve in larger networks, and does it remain informative at scale?

- Could the authors provide a clear diagram showing the inputs, outputs, and how the motif regularization integrates into the training pipeline?

- What can simpler network statistics, such as degree distribution, clustering coefficient, and assortativity, tell us about the network that motifs cannot? What do motifs reveal that these metrics miss?

- Does the framework extend naturally to hypergraph settings where higher-order interactions are explicit?

- Could the authors clarify the precise relationship between the weight matrix W and  W̃,?

-Minor: spacing between sentences; refer all figures

**Limitations:**

yes

**Strengths And Weaknesses:**

**Soundness:** It is a well-written paper with clear motivation. The theoretical stability analysis is clean and correctly grounded in the Hartman–Grobman theorem, and the experimental results across multiple tasks consistently support the proposed hierarchy. However, the stability conditions are sufficient but not necessary, and it remains unclear how well the motif-level analysis scales to the high-dimensional, non-linear regime of full networks.

**Presentation:** The paper is clearly written and well-structured.  I enjoyed reading it.

**Significance:** The motif-based regularization framework offers a principled and biologically motivated tool for designing RNNs with targeted computational properties, which is a practically useful contribution. The robustness-flexibility trade-off framing is intuitive and likely to be of interest to both the ML and computational neuroscience communities.

**Originality:**  The unification of all 13 three-node motifs under a single differentiable regularization framework, with a stability-grounded hierarchy, is a novel and well-motivated contribution.

---

> ### Author Rebuttal · Authors · 2026-03-31
>
> We sincerely thank the reviewer for the constructive feedback. We especially appreciate the recognition that the paper is well motivated and that the results consistently support the proposed hierarchy. We also appreciate the broader perspective on how motif-level structure should be interpreted in the high-dimensional nonlinear regime of full networks.
>
> ## Q1. Can local stability properties derived from isolated three-node motifs persist in a large, densely overlapping RNN?
> ---
> We agree that the hierarchy is derived from isolated three-node motifs, whereas in a large fully connected RNN these motifs are densely overlapping. We therefore do not claim strict one-to-one persistence of local motif stability at the full-network level. Rather, enriching different motif levels provides a structural bias that shifts the collective regime. This is supported by both task-level results and the paper’s dynamical analyses, which consistently show a shift from more convergent Level-1 dynamics to more expansive Level-3 dynamics. We will clarify this scope more explicitly in the revision.
>
> ## Q2. Why focus on three-node motifs, and could four-node motifs reveal qualitatively different regimes?
> ---
> The framework itself is not restricted to three-node motifs: the same counting and regularization idea can in principle extend to larger or variable-sized motifs. We focus on the three-node case because full enumeration is still manageable there, which makes the structure-dynamics connection easier to analyze and interpret. Larger motifs are a natural next step beyond the present paper.
>
> ## Q3. How does motif distribution evolve at scale, and does it remain informative?
> ---
> We interpret this as asking whether both parts of our framework remain valid in larger networks: (1) the ability to constrain the motif distribution to a target distribution, and (2) the dynamical meaning of the L1/L2/L3 hierarchy.
>
> For motif distribution control, the answer is yes empirically. In additional tests on networks with 256, 512, 600, 700, and 800 neurons, the regularizer consistently steers the network toward the desired motif distribution. For the L1/L2/L3 hierarchy itself, we are currently running larger-scale experiments. Since 512 is not a specially tuned scale for our method, and the results there are already strong, we expect the same qualitative hierarchy to remain informative in larger networks as well. We are working to add these larger-scale results to the anonymous repository soon:
>
> https://anonymous.4open.science/r/Functional-building-blocks-of-neural-networks--5F5E/results/large_scale_motif_analysis
>
> ## Q4. Could the authors provide a clear diagram showing the inputs, outputs, and how the motif regularization integrates into the training pipeline?
> ---
> Yes. We will add a clearer end-to-end training diagram. PDM takes a 1-d input and produces a 1-d output. Sequential MNIST takes a 28-step sequence with 28-d input at each step and outputs a 10-class digit prediction. TIDIGITS takes MFCC features reshaped into a 20 x 20 representation and outputs a 10-class spoken-digit prediction. IP-v2 takes a 4-d input and outputs a 1-d action. Ant-v2 takes a 27-d input and outputs an 8-d action. In all cases, training uses $L_{total} = L_{task} + \lambda * L_{motif}$, where the motif term biases recurrent connectivity toward the target motif distribution.
>
> ## Q5. What do motifs reveal that simpler statistics such as degree distribution, clustering coefficient, and assortativity miss?
> ---
> Simpler statistics such as degree distribution, clustering coefficient, and assortativity summarize node-level or global properties of a network, but they do not explicitly distinguish the local recurrent wiring patterns central to our analysis. Two networks may have similar degree or clustering statistics while containing very different proportions of unidirectional, reciprocal, or cyclic motifs. Motifs therefore provide a more fine-grained descriptor aligned with local dynamical regimes.
>
> ## Q6. Does the framework extend naturally to hypergraph settings where higher-order interactions are explicit?
> ---
> Our current framework defines higher-order structure through motifs induced by pairwise recurrent connectivity, whereas hypergraphs treat higher-order interactions as explicit primitives. Since hypergraphs already have established mathematical representations, for example through incidence matrices, we therefore view this as an interesting and promising direction for future work.
>
> ## Q7. Could the authors provide a clearer training pipeline and clarify the relation between W and $\tilde{W}$?
> ---
> The distinction is now made explicit: W is the real recurrent weight matrix of size N x N, used in forward dynamics and optimized by gradient descent, while $\tilde{W} = \mathrm{sigmoid}(\beta ((W \odot W)- \theta))$ is the differentiable relaxed adjacency proxy used only for motif counting and regularization.

---

> > ### Author Rebuttal · Reviewer_HLFU · 2026-04-02
> >
> > I would like to thank the authors for addressing all my comments.

---

### Official Review · Reviewer_Cah1 · 2026-03-13

**Soundness:** 3
**Presentation:** 3
**Significance:** 3
**Originality:** 3
**Overall Recommendation:** 5
**Confidence:** 4

**Summary:**

This paper studies whether small three-node connection patterns, or motifs, can serve as functional building blocks for recurrent neural networks. The authors analyze the 13 directed three-node motifs by approximating their dynamics near equilibrium and grouping them into three stability levels. Intuitively, Level 1 motifs are the most stabilizing, Level 2 motifs impose stronger constraints on simple feedback loops, and Level 3 motifs contain richer feedback and are the least stable. The paper then introduces a motif-based regularization method that biases an RNN toward desired motif frequencies using exact motif counting together with a differentiable approximation. Experiments suggest that networks biased toward more stable motifs are more robust to input noise on prediction tasks, while networks biased toward less stable, feedback-rich motifs learn faster and explore better on reinforcement-learning control tasks.

**Compliance With Llm Reviewing Policy:**

Affirmed.

**Final Justification:**

I updated the scores based on the rebuttal by the authors to my comments and those of the other reviewers.

**Key Questions For Authors:**

How were the hyperparameters selected for each task, especially the motif-regularization terms? An ablation or sweep over the key regularization parameters would materially affect my confidence in the reported gains.
What exact reinforcement-learning algorithm and protocol were used, and were the tasks modified in any way to make recurrence necessary? This matters for interpreting the RL gains.
How do the results compare to stronger baselines such as GRU/LSTM or standard regularization methods aimed at stability or robustness?
What do the error bars represent, and how many random seeds were used for training and evaluation? Please also clarify why some vanilla-RNN curves appear to lack uncertainty bands.
How is MNIST fed to the recurrent network, and can the appendix include the motif-regularization loss over training epochs to show that the intended structural bias is actually maintained?

**Limitations:**

No. The paper should add an explicit limitations section discussing the narrow baseline set, incomplete hyperparameter reporting, lack of a public code link, and uncertainty about how well the findings transfer beyond vanilla RNNs.

**Strengths And Weaknesses:**

Strengths
The paper makes an interesting connection between local connectivity patterns, dynamical-systems analysis, and downstream network behavior.
The central idea is easy to understand at a high level: different small feedback patterns may bias an RNN toward stability or flexibility.
The three-level motif hierarchy gives the paper a clear conceptual structure.
The overall empirical story is coherent: more stable motifs appear to help noise robustness, while less stable motifs appear to help adaptive control.
The paper is generally well written and the motif-regularization idea could be useful beyond the specific experiments shown here.
Weaknesses
Hyperparameter selection is not explained well enough. In particular, the motif-regularization parameters need clearer reporting, and performance with and without the regularizer should likely be tuned separately.
The paper would benefit from an explicit ablation over the key regularization parameters. At present it is hard to tell how sensitive the results are to the chosen motif-loss settings.
The reinforcement-learning setup is underspecified. The paper does not clearly explain the training protocol in enough detail, and it is also unclear whether the tasks used here truly require recurrence or whether simpler non-recurrent solutions would already perform well.
Baselines are limited. The comparison is mainly against a vanilla RNN, whereas stronger recurrent baselines such as GRU/LSTM or simpler regularization baselines aimed at improving stability and robustness would make the empirical case more convincing.
Uncertainty reporting needs clarification. The paper does not define the error bars clearly enough, and it is not obvious how many seeds were used or why some vanilla-RNN curves appear without visible uncertainty bands.
A few experimental details remain unclear, such as how MNIST is presented sequentially to the recurrent network and whether the motif-regularization term remains active and stable throughout training.
I did not find a code release link, which limits reproducibility.

---

> ### Author Rebuttal · Authors · 2026-03-31
>
> **Anonymous code repository**
> https://anonymous.4open.science/r/Functional-building-blocks-of-neural-networks--5F5E
>
> ## Q1. How were the hyperparameters selected for each task, especially the motif-regularization terms?
> ---
> Our objective is:
>
> $L_{\mathrm{total}} = L_{\mathrm{task}} + \lambda L_{\mathrm{motif}}$
>
> The structural branch uses the relaxed adjacency proxy:
>
> $\tilde{W} = \mathrm{sigmoid}(\beta ((W \odot W)- \theta))$
>
> Here, $\lambda$ controls the task-structure trade-off, while $\beta$ and $\theta$ control the sharpness and threshold of the relaxation.
>
> In the revised version, vanilla RNNs and motif-regularized RNNs are tuned independently under the same validation budget. We also add dedicated sensitivity analyses. Specifically, we scan $\lambda \in \{1, 10^2, 10^4, 10^6, 10^8\}$ and target frequency $\in \{0.1, 0.3, 0.5, 0.7, 0.9\}$, and separately scan slope $\in \{10, 30, 10^2, 3 \times 10^2, 10^3\}$ and bias $\in \{10^{-4}, 10^{-3}, 10^{-2}, 5 \times 10^{-2}, 10^{-1}\}$. Across the lambda and target-frequency scan, task accuracy remains high while motif matching increases smoothly with stronger motif regularization. Across the slope/bias scan, strong performance persists for slope $10^2$--$10^3$ and bias $10^{-2}$--$5 \times 10^{-2}$. The defaults ($\lambda = 10^6$, target frequency $= 0.3$, slope $= 10^2$, bias $= 5 \times 10^{-2}$) lie inside this broad stable regime rather than at a brittle edge case.
>
> ## Q2. What exact reinforcement-learning algorithm and protocol were used, and were the tasks modified in any way to make recurrence necessary?
> ---
> We agree that the RL setup was under-specified. The RL experiments use PPO with recurrent actor and critic networks. The main settings are: rollout horizon 2048, discount factor $\gamma = 0.98$, GAE $= 0.98$, clip coefficient $\epsilon = 0.2$, optimizer Adam, and critic weight decay 0.001.
>
> We do not claim that these benchmarks inherently require recurrence. Rather, recurrence is used as a controlled backbone so that different motif priors can be compared on the same recurrent substrate.
>
> ## Q3. How do the results compare to stronger baselines such as GRU/LSTM or standard regularization methods aimed at stability or robustness?
> ---
> We agree that stronger baselines improve the empirical case. We therefore added LSTM, GRU, Orthogonal Initialization, Low-rank constraints, and Spectral Normalization.
>
> The main pattern remains task-dependent. On PDM, Level1-motif1 is best ($91.9 \pm 1.4$), outperforming Vanilla ($87.1 \pm 1.7$), GRU ($86.5 \pm 7.2$), LowRank-r128 ($87.3 \pm 5.3$), and SpectralNorm ($83.0 \pm 4.6$). On Sequential MNIST, LSTM is strongest ($68.19 \pm 1.51$), while motif1 remains competitive ($67.0 \pm 3.4$). On TIDIGITS, motif1 again performs best ($45.0 \pm 1.9$), ahead of Vanilla ($40.9 \pm 2.5$).
>
> For RL, the same separation remains visible. On Ant, Level3-motif12 is strongest at both 2M and 5M steps ($2725.0 \pm 233.5$ and $4258.0 \pm 166.6$), exceeding Vanilla ($1961.1 \pm 133.8$ and $2882.1 \pm 243.9$) and the structural baselines. On Inverted Pendulum, both LSTM and motif12 reach ceiling performance. Thus, stronger baselines strengthen the comparison, but they do not erase the task-dependent motif hierarchy predicted by our stability framework.
>
> ## Q4. What do the error bars represent, and how many random seeds were used for training and evaluation?
> ---
> All quantitative results are aggregated over 10 independent random seeds. In tables, we report mean $\pm$ standard deviation; in the statistical tests in the text, we report standard error. We will make this distinction explicit in the revised captions and appendix, and we will include Vanilla-RNN error bars in the revised figures.
>
> ## Q5. How is MNIST fed to the recurrent network, and can the appendix include the motif-regularization loss over training epochs to show that the intended structural bias is actually maintained?
> ---
> Yes. In Sequential MNIST, each image is reshaped into a 28-step sequence, where each step is a 28-dimensional row vector. We will state this explicitly in the experimental setup.
>
> We also consider a two-stage training setting, where the motif objective is optimized first and task learning is performed afterward. After motif optimization, both motif frequency and motif loss remain essentially unchanged during task training, as shown in the anonymous figure below:
>
> https://anonymous.4open.science/r/Functional-building-blocks-of-neural-networks--5F5E/docs/figures/fig-a3-1.png
>
> ## Response to Weakness
> ---
> We appreciate the reviewer’s emphasis on empirical rigor. The main weaknesses of the original submission were incomplete hyperparameter reporting, a narrow baseline set, and insufficient protocol detail. We addressed these by expanding the comparisons, clarifying the training procedures, unifying uncertainty reporting, and releasing anonymous code.

---

> > ### Author Rebuttal · Reviewer_Cah1 · 2026-04-04
> >
> > I would like to thank the authors for their rebuttals, which addressed the concerns I had. As such, I will increase my score accordingly.

---

### Decision · Program_Chairs · 2026-04-30

**Decision:**

Accept (regular)

**Comment:**

This work is a theoretical exploration of how small scale (3-node) subnetworks can influence global activity patterns in RNNs. Systematic understanding of what drives RNNs into different operating regimes and guide their properties is of significant interest to the ICML community and a very timely and important topic. The reviewers noted that the work was very clear and easy to understand, which is not always the case for theory work. There were some concerns, however most were address in rebuttal. The remaining concerns, such as the application to LLMs or extension to larger sub-motifs, while interesting seem to be beyond the scope of this work. As all reviewers were positive (note the reviewer who cited a "3" score did indicate their concerns were addressed) I am recommending this work be accepted.